# Single-cell RNA-sequencing uncovers transcriptional states and fate decisions in haematopoiesis

Emmanouil I. Athanasiadis [1,2,3], Jan G. Botthof[1,2,3], Helena Andres[4], Lauren Ferreira[1,2,3,5], Pietro Lio[4] & Ana Cvejic[1,2,3]

The success of marker-based approaches for dissecting haematopoiesis in mouse and human is reliant on the presence of well-defined cell surface markers specific for diverse progenitor populations. An inherent problem with this approach is that the presence of specific cell surface markers does not directly reflect the transcriptional state of a cell. Here, we used a marker-free approach to computationally reconstruct the blood lineage tree in zebrafish and order cells along their differentiation trajectory, based on their global transcriptional differences. Within the population of transcriptionally similar stem and progenitor cells, our analysis reveals considerable cell-to-cell differences in their probability to transition to another committed state. Once fate decision is executed, the suppression of transcription of ribosomal genes and upregulation of lineage-specific factors coordinately controls lineage differentiation. Evolutionary analysis further demonstrates that this haematopoietic programme is highly conserved between zebrafish and higher vertebrates.

[1] Department of Haematology, University of Cambridge, Cambridge, CB2 0XY, UK. [2] Wellcome Trust Sanger Institute, Wellcome Trust Genome Campus, Cambridge, CB10 1SA, UK. [3] Wellcome Trust – Medical Research Council Cambridge Stem Cell Institute, Cambridge, CB2 1QR, UK. [4] Computer Laboratory, University of Cambridge, Cambridge, CB3 0FD, UK. [5]Present address: Biotechnology Innovation Centre, Rhodes University, Grahamstown, 6139, South Africa. Emmanouil I. Athanasiadis and Jan G. Botthof contributed equally to this work. Correspondence and requests for materials should be addressed to A.C. (email: as889@cam.ac.uk)

Mammalian blood formation is the most intensely studied system of stem cell biology, with the ultimate aim to obtain a comprehensive understanding of the molecular mechanisms controlling fate-determining events. A single cell type, the haematopoietic stem cell (HSC), is responsible for generating more than 10 different blood cell types throughout the lifetime of an organism[1]. This diversity in the lineage output of HSCs is traditionally presented as a stepwise progression of distinct, transcriptionally homogeneous populations of cells along a hierarchical differentiation tree[2–6]. However, most of the data used to explain the molecular basis of lineage differentiation and commitment were derived from populations of cells isolated based on well-defined cell surface markers[7]. One drawback of this approach is that a limited number of markers are used simultaneously to define the blood cell identity. Consequently, only a subpopulation of the overall cellular pool is examined and isolated cells, although homogeneous for the selected markers, show considerable transcriptional and functional heterogeneity[8–12]. This led to the development of various refined sorting strategies in which new combinations of marker genes were considered to better 'match' the transcriptional and functional properties of the cells of interest.

The traditional model of haematopoiesis assumes a stepwise set of binary choices with early and irreversible segregation of lymphoid and myeloid differentiation pathways[2, 3]. However, the identification of lymphoid-primed multipotent progenitors[4], which have granulocytic, monocytic and lymphoid potential, but low potential to form megakaryocyte and erythroid lineages prompted development of alternative models of haematopoiesis. More recently, it has been demonstrated that megakaryocyte–erythroid progenitors can progress directly from HSC without going through a common myeloid intermediate (CMP)[13]; or that the stem cell compartment is multipotent, while the progenitors are unipotent[6]. Clear consensus on the lineage branching map, however, is still lacking.

Recent advances in single-cell transcriptional methods have made it possible to investigate cellular states and their transitions during differentiation, allowing elucidation of cell fate decision mechanisms in greater detail. Computational ordering methods have proved to be particularly useful in reconstructing the differentiation process based on the transcriptional changes of cells at different stages of lineage progression[14–16].

Here we create a comprehensive atlas of single-cell gene expression in adult zebrafish blood cells and computationally reconstructed the blood lineage tree in vivo. Conceptually, our approach differs from the marker-based method in that the identity of the cell type/state is determined in an unbiased way, i.e., without prior knowledge of surface markers. The transcriptome of each cell is projected on the reconstructed differentiation path giving complete insight into the cell state transitions occurring during blood differentiation. Importantly, development of this strategy allowed us, for the first time, to asses haematopoiesis in a vertebrate species in which surface marker genes/antibodies are not readily available. Finally, this study provides unique insight into the regulation of haematopoiesis in zebrafish and also, along with complementary data from mouse and human, addresses the question of interspecies similarities of haematopoiesis in vertebrates.

## Results

### Single-cell RNA-sequencing of zebrafish haematopoietic cells.
As an alternative to marker-based cellular dissection of haematopoietic hierarchy, we have set out to classify haematopoietic cells based on their unique transcriptional state. We started by combining FACS index sorting with single-cell RNA-seq to reveal the cellular properties and gene expression of a large number of blood cells simultaneously. To cover the entire differentiation continuum, kidney-derived blood cells from eight different zebrafish transgenic reporter lines and one non-transgenic line were FACS sorted (Fig. 1a and Supplementary Table 1). Each blood cell was collected in a single well of a 96-well plate. At the same time, information about the cell size (FSC) and granularity (SSC), as well as the level of the fluorescence, were recorded.

RNA from each cell was isolated and used to construct a single mRNA-seq library per cell, which was then sequenced to a depth of around $1 \times 10^6$ reads per library. Following quality control (QC), 1422 cells were used for further analysis and for benchmarking of different alignment methods (Supplementary Figs. 1, 2 and 3). Importantly, the average single-cell profiles showed good correlation with independent bulk samples (PCC = 0.7–0.9, Supplementary Fig. 3e). In addition, PCA, ICA and diffusion maps (Supplementary Fig. 4a) showed that cells were intermixed irrespective of the fish or the plate they originated from. This confirmed that the cells were separated in the analyses based on their biological differences rather than batch-induced biases.

**HSPC fates through a single path in the state space.** A dynamic repertoire of gene expression in thousands of cells during differentiation could be used to infer a single branched differentiation trajectory. Due to the unsynchronised nature of haematopoiesis, each single cell exhibits a different degree of differentiation along the differentiation continuum. Therefore, the generated trajectory could be used to infer the differentiation path of a single cell. To examine the transcriptional transition undergone by differentiating cells, we identified the 1845 most highly variable genes (Supplementary Fig. 4b) and performed expression-based ordering using Monocle2[15]. Based on global gene expression profiles of the cells, we identified five (1–5) distinct cell 'states' (Fig. 1b). To ensure the robustness of this approach, we verified computationally that changes in the highly variable genes and Monocle2 settings only lead to minor differences in the trajectory, mainly around the branching points (Supplementary Fig. 5).

Differential expression analysis of each state vs. all other states, followed by gene ontology (GO) enrichment analysis (see 'Methods' section), provided clear insights into the cell types in each state (Fig. 1c). Specifically, state 1 contains GO terms relating to antigen processing, including genes that are highly expressed in the monocyte lineage, such as *cd74a/b*[17], *ctss2.2*[18] and *mhc2dab*[19] (Supplementary Data 1). The functionality of state 2 relates to leucocyte migration, including genes specific to neutrophils, e.g., *cxcr4b*[20], *rac2*[21] and *wasb*[22, 23] (Supplementary Data 1). State 3 is highly enriched for genes that are involved in ribosome biogenesis, including *fbl* (Fibrillarin) and *pes* (Pescadilo), both of which are critical for stem cell survival[24, 25] (Supplementary Data 1). Since there is also enrichment for HSC homoeostasis, this state is most likely to be haematopoietic stem/progenitor cells (HSPCs). With GO terms that include gas exchange and erythrocyte differentiation involving the adult haemoglobins, *ba1*, *ba1l* and *hbaa1*[26] together with the erythroid-specific aquaporin gene, *aqp1a*[26, 27] (Supplementary Data 1), state 4 can be assigned to the erythroid lineage. Finally, state 5 has functionality that is relevant for circulatory system development and blood coagulation, both of which include *itga2b* (also known as *cd41*) together with its heterodimer *itgb3b*[28] (Supplementary Data 1). Since these gene lists include other genes that interact with this platelet integrin receptor complex, as well as additional genes relevant for platelet function, we assigned this cell state to thrombocytes. Mature lymphocytes could not be detected, most

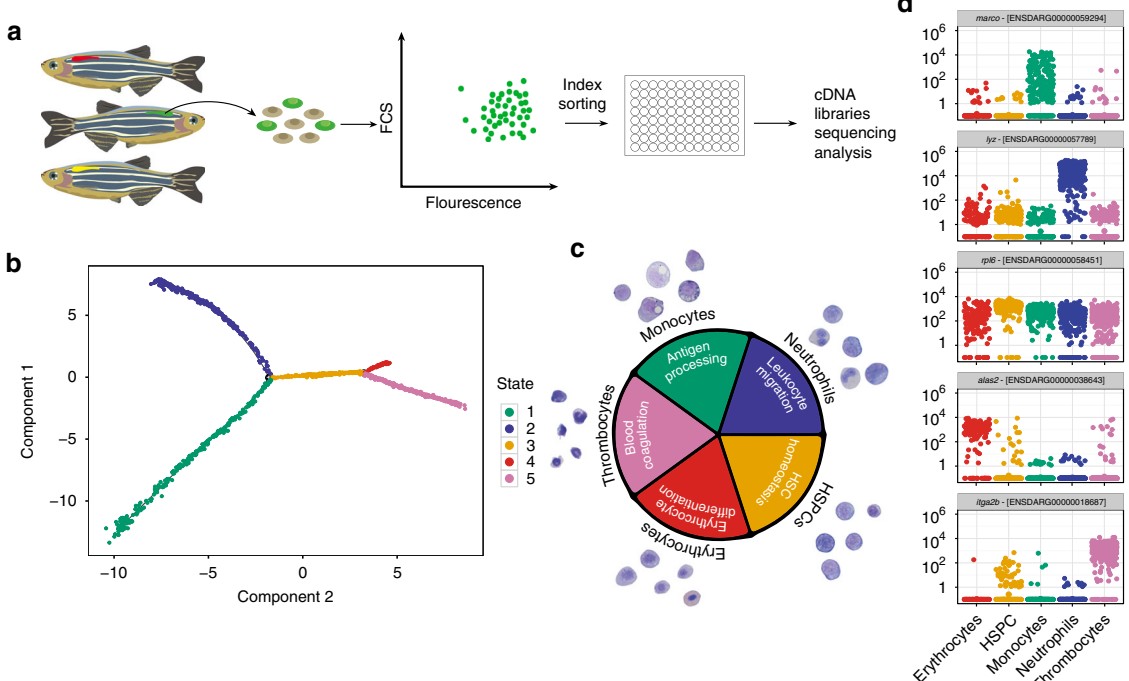

**Fig. 1** Pseudotime ordering reveals a gradual transition of cells from immature to more differentiated within the myeloid branch. **a** Experimental strategy for sorting single cells from transgenic zebrafish lines. Cells were collected from a single kidney of each line and sorted for expression of the fluorescent transgene. Index sorting was used to dispense single cells into a 96-well plate and these were subsequently processed for RNA-seq analyses. **b** Five cell states were predicted using the Monocle2 algorithm for temporal analyses of single-cell transcriptomes. **c** Analysis of genes that are differentially expressed across the five states (given the same colour code used in **b**) reveals GO terms (inner circle) that are highly pertinent to specific cell types. The outer circle shows examples of May–Grünwald Giemsa-stained cells from kidneys of transgenic lines that largely label each particular cell type. **d** Jitter plots showing the expression (y axis) of differentially expressed marker genes in each cell type (x axis). Each dot in the jitter plot shows the expression of the gene $\log_{10}$ (counts +1) in each cell

likely as T cells mature in the thymus and B cells are comparatively rare and were not enriched for.

To experimentally confirm our computational predictions, we sorted cells from transgenic lines that were the most abundant in each of the five states (Fig. 2) and stained them using May–Grünwald Giemsa staining. Indeed, the morphological properties of the sorted cells (Fig. 1c and Supplementary Figs. 6–7) matched the assigned cell types, therefore adding confidence to these cell-type assignations. As expected, the signature genes, such as *marco*, *lyzC*, *hhex*, *alas2* and *itga2b*, were within the most differentially expressed genes in monocytes, neutrophils, HSPC, erythrocytes and thrombocytes, respectively (Fig. 1d).

Taken together, the reconstructed branched tree revealed a gradual transition of myeloid cells from immature to more differentiated cells. Within this tree, HSPCs assumed a committed state through a single path, suggesting that during steady state haematopoiesis, HSPCs can reach a specific cell fate through only one type of intermediate progenitor.

**Distinct state cells with different repopulation potential.** Functional in vivo transplantation assays have been traditionally used to assess the differentiation potential of different haematopoietic populations. To examine the repopulation and lineage potential of the cells within different states, we sorted cells from *Tg(mpx:EGFP)*[29], *Tg(gata1:EGFP)*[30] and *Tg(runx1:mCherry)*[31] fish to enrich for neutrophil, erythroid and HSPC cell state, respectively. We next injected 500 donor cells into sublethally irradiated, immunocompromised *rag2*[E450fs−/−] zebrafish[32] and assessed their engraftment at 1 day, 4- and 14 weeks post injection (PI) (Fig. 3a).

Analysis of kidney repopulation revealed that *mpx+*, *gata1+* and *runx1+* cells were able to home to the kidney 1 day PI (Fig. 3b). However, only progeny of *runx1+* cells were detectable at 4 weeks PI in all examined recipients (Fig. 3b). No progeny of *mpx+* and *gata1+* cells were evident at the same time point. To examine the lineage output of *runx1+* cells following transplantation, we sorted engrafted *runx1+* kidney cells 4 and 14 weeks PI and processed them for scRNA-seq analysis. The scRNA-seq data from 302 engrafted *runx1+* cells projected onto a Monocle trajectory revealed the multilineage potential of donor *runx1* cells at both 4 and 14 weeks PI (Fig. 3c). These data strongly suggested that at least some of these cells were HSCs.

According to transplantation assays, cytospins and transcriptional profiling of cells prior and following transplantation, cells located in the branches of the Monocle tree show progression of lineage-restricted progenitors to mature blood cells with no repopulation potential. However, cells in the middle of the Monocle tree (state 3) are a mixture of progenitors and HSCs with long-term multilineage potential.

**Heterogeneity within the HSPC branch of the lineage tree.** To increase the number of HSCs in our data set and the resolution in the HSPC branch of the Monocle trajectory, we added the 302 transplanted *runx1+* cells to our 1422 previously sequenced cells. We re-analysed the whole data set (1724 cells in total), and generated a new Monocle trajectory (Fig. 4a).

Next, we considered the frequency of potential HSCs in this data set. To do so, we computed the stemness $S^{rel}$ index[33], using the Kullback–Leibler distance of the predicted probabilities compared to the expected one, for each of the four different

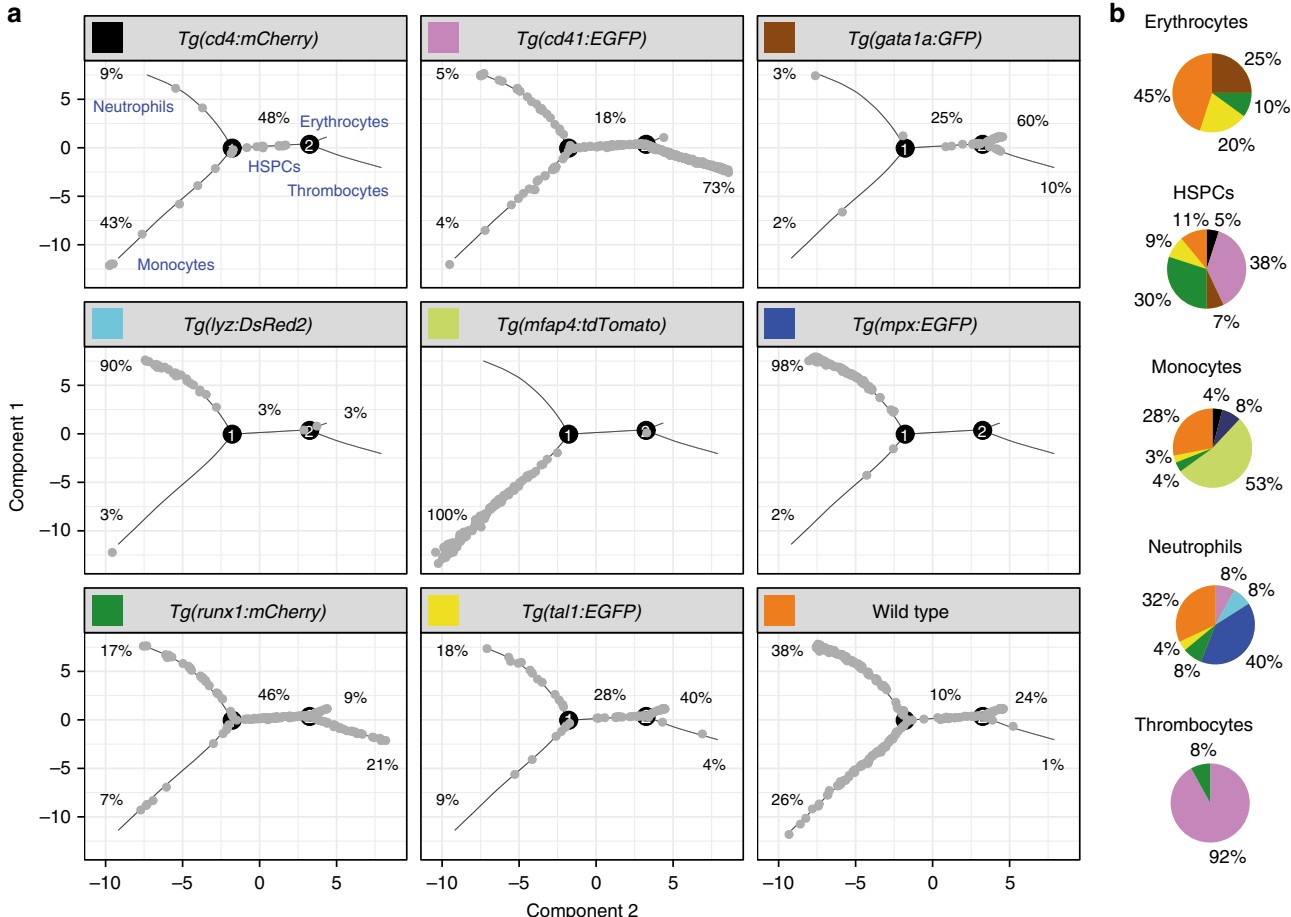

**Fig. 2** The distribution of cells from different transgenic lines modelled by Monocle. **a** The trajectories of cell states predicted by Monocle are shown in grey for each transgenic line used, with the associated cell types labelled in blue. The percentage of cells from each transgenic line contributing to each state is given next to the relevant trajectory. **b** Pie charts showing the contribution of transgenic lines to each cell type. The colour code relates to the colours given in the headers for each transgenic line used in **a**

branches (Fig. 4a, b). The lower the 'stemness' factor, the higher the confidence that a particular cell is a stem cell. Using the threshold of 3 sigma over the mean stemness value (0.05), our analysis predicted that 35 out of 214 cells in the middle part of the tree are potential HSCs. The majority of cells that were identified as stem cells originated from the *cd41* (13 cells) and *runx1* (14 cells) transgenic lines (Fig. 4c). It should be noted that both these lines have been previously identified to contain transplantable HSCs[31, 34], lending further confidence to our computational prediction. This suggests that, although both stem and progenitor cells are intermixed on the trajectory due to their overall similar transcriptomes, their lineage potentials (and thus stemness scores) are distinct.

**Ribosomal genes and lineage factors control differentiation.** Differentiation generally involves specific regulated changes in gene expression. To understand the dynamics of transcriptional changes during the differentiation of myeloid cells, we examined trends in gene expression in each of the four branches (Fig. 5). Dynamically expressed genes within each of the branches showed two main trends (see 'Methods' section). These included genes gradually upregulated through pseudotime and genes gradually downregulated (Fig. 5a, b).

Genes upregulated in pseudotime included well known genes related to the specific function of the relevant cell type (Fig. 5b). The majority of cells characterised as erythroid dynamically expressed genes such as *alas2, aqp1a.1, ba1, ba1l, cahz* and *hbaa1*.

Similarly, cells in the monocyte branch dynamically expressed genes like *c1qa, cd74a, ifngr1, marco, myod1* and *spi1a*; among other genes, the *cebpb, cfl1, cxcr4b, illr4, mpx* and *ncf1* were upregulated in pseudotime in the neutrophil branch and thrombocytes dynamically expressed *fn1b, gp1bb, itga2b, mpl, pbx1a* and *thbs1b*. A complete list of all genes that were dynamically expressed across pseudotime can be found in Supplementary Data 1.

Interestingly, genes downregulated through pseudotime (Fig. 5b) in each of the four branches were consistently enriched for genes involved in ribosome biosynthesis, as revealed by GO terms 'biosynthetic process', 'ribosome' and 'translation' (Supplementary Data 1). This is an interesting finding, because previous studies suggested that HSCs have significantly lower rates of protein synthesis than other haematopoietic cells[35]. Therefore, we went on to investigate the expression of ribosomal proteins in pseudotime in greater depth (Fig. 5c).

Out of 168 genes annotated as 'ribosomal proteins' on Ensembl BioMart database (Supplementary Data 1), 89 genes had low, random expression in our data set (Fig. 5c). These genes encoded mainly mitochondrial ribosomal proteins (Fig. 5c). In contrast, 79 genes that showed high expression across all cells encoded cytoplasmic ribosomal proteins and were downregulated in pseudotime in all four branches (Fig. 5c). Importantly, the observed downregulation of ribosomal genes in pseudotime was not correlated with the cell cycle state of the cell, apart from a weak correlation in the erythrocytic lineage (Supplementary

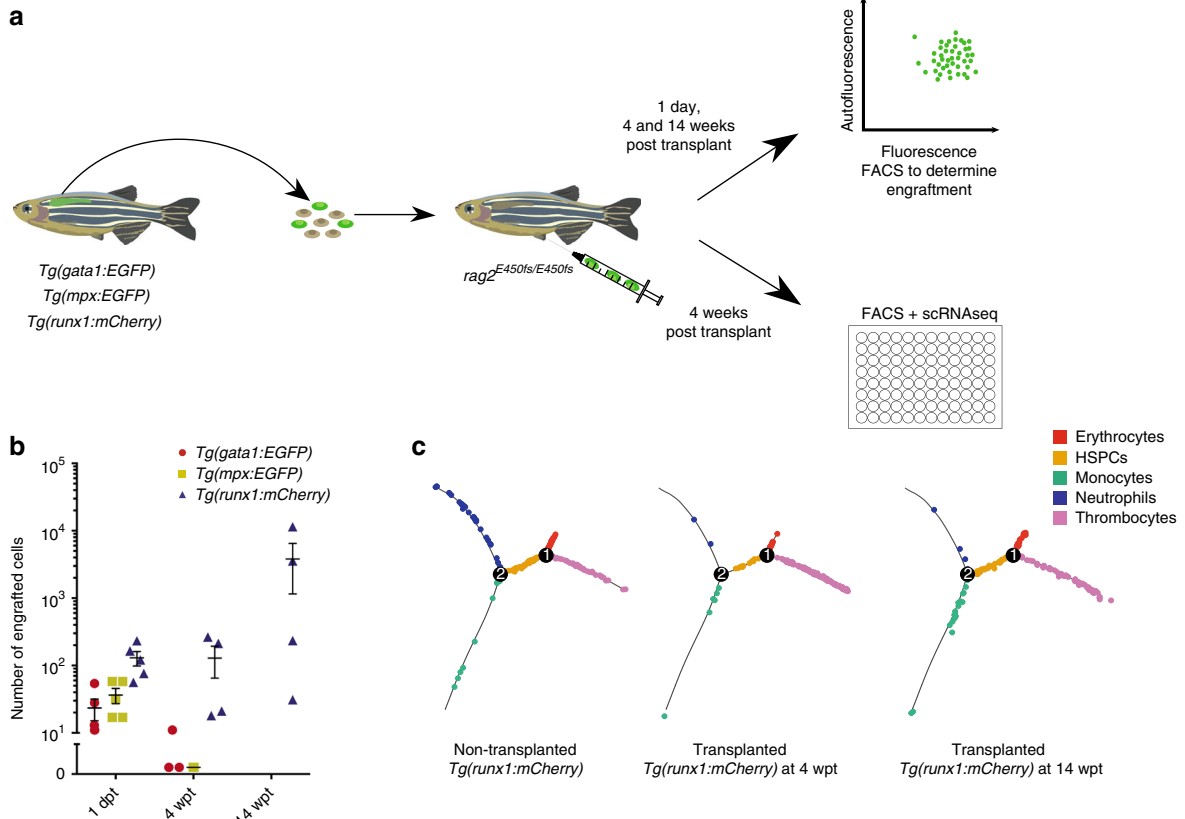

**Fig. 3** Cells within distinct states have different repopulation potentials. **a** Experimental strategy for the adult transplantation experiment. Kidneys were dissected from transgenic donor fish and sorted for cells expressing the fluorescent transgene. Positive cells were collected and injected into sublethally irradiated *rag2^E450fs−/−* fish. **b** Assessment for engraftment was made 1 day, 4- and 14 weeks post transplantation using flow cytometry. Successfully engrafted fluorescent donor cells were isolated at 4 weeks PI by index sorting single cells into a microtitre plate for subsequent RNA-seq analyses. **c** Distribution of *runx1*+ cells, from non-transplanted (left) and transplanted fish at 4 (middle) and 14 wpt (right), modelled by Monocle

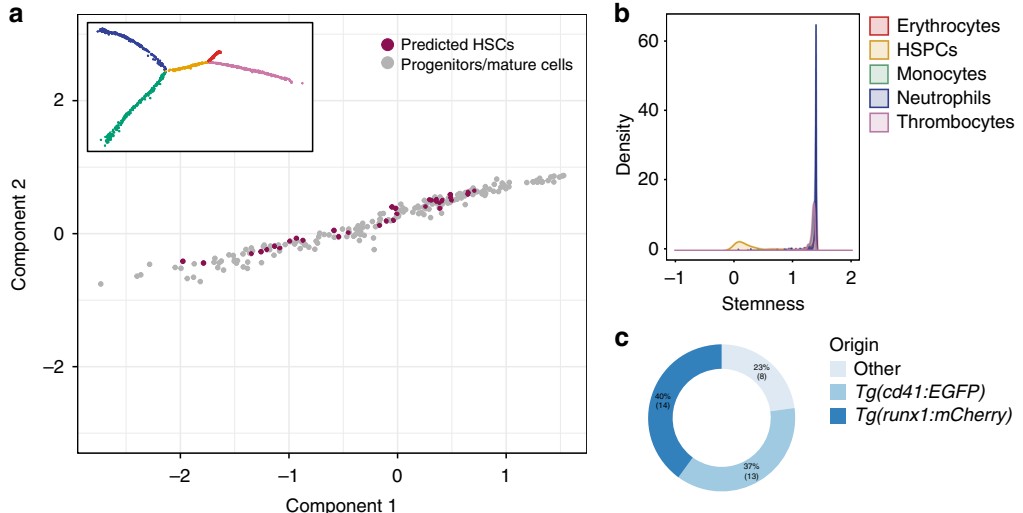

**Fig. 4** Transcriptionally similar cells display different probabilities of being stem cells. **a** Cells predicted to be stem cells in the middle part of the lineage tree according to their stemness index. The insert shows the new Monocle tree including transplanted cells (1724 single cells and 1871 highly variable genes). **b** Distribution of stemness scores in different branches of the tree showing the presence of potential HSCs exclusively in the HSPC branch. **c** Contribution of different transgenic lines to predicted stem cells

Fig. 8). These findings further indicate that there is a common developmental event in which suppression of transcription of ribosomal genes and upregulation of lineage-specific factors direct lineage commitment and terminal differentiation.

Next, we compared ribosomal gene expression between the predicted HSCs and the remaining progenitors in the middle branch. The absolute number of ribosomal transcript was similar between the two populations (CV score, $0.180 \pm 0.033$). In

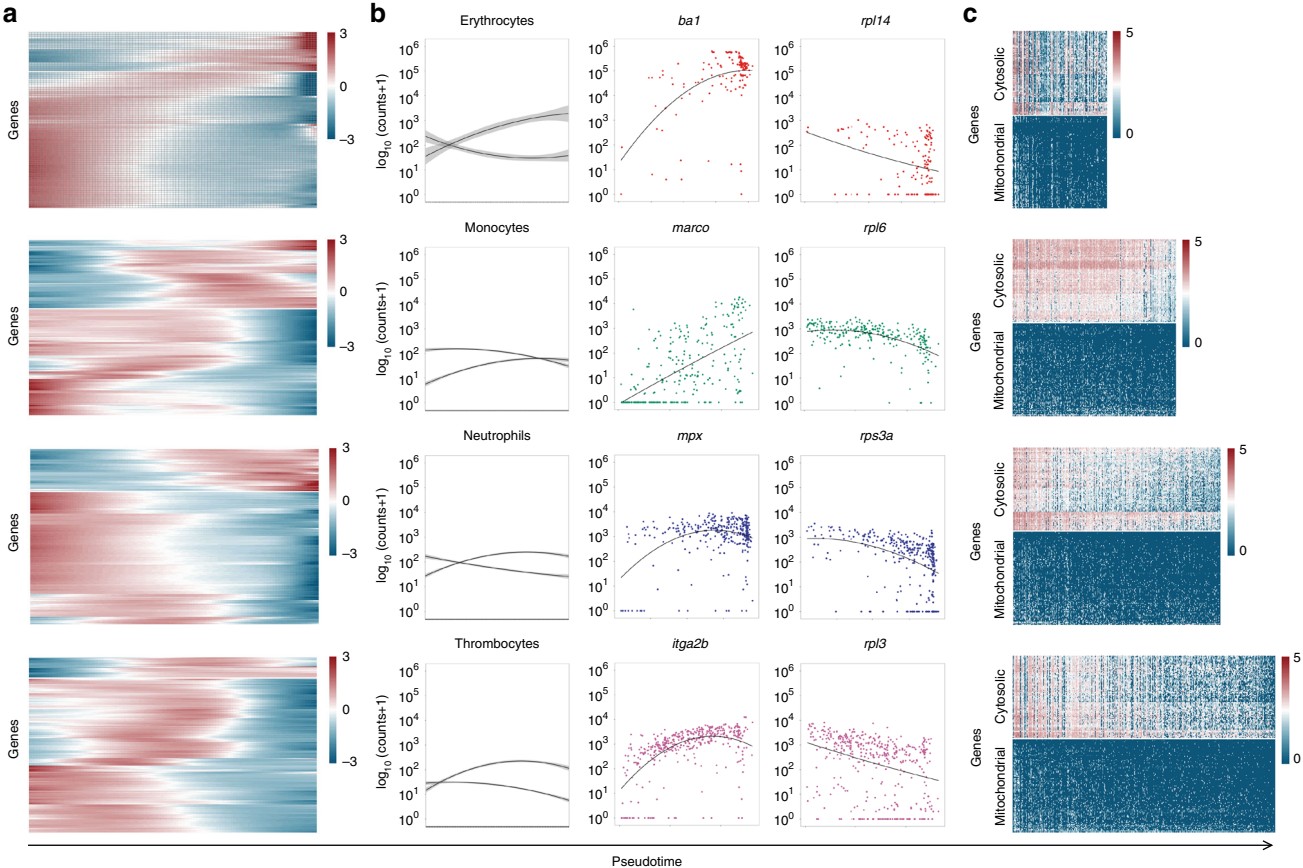

**Fig. 5** Lineage differentiation is defined by two main trends in gene expression. **a** Heatmap of genes whose expression changed dynamically during pseudotime in each of the four branches. **b** Graph showing the average expression pattern of the dynamically expressed genes that follow the same trend across pseudotime. For each of the cell states, one gene is presented that follows one of the two main trends. Standard error is shown as a grey area around the trend lines. **c** Heatmap of expression of 168 genes annotated as 'ribosomal proteins' genes in pseudotime in each of the four branches

agreement with this, there was a high Pearson correlation (0.986) between predicted HSCs and progenitors. Overall, this suggests highly similar ribosomal gene expression between HSCs and more committed progenitors. However, it is possible that modest differences in ribosomal gene expression between HSCs and progenitors cannot be detected by the methodologies currently available.

In order to address whether this trend has been evolutionarily conserved from zebrafish to mammals, we considered the correlation in ribosomal gene expression between human phenotypic HSCs (CD34+ CD38− CD45RA− CD90+ CD49f+) and the different progenitor fractions (for details, please see 'Methods' section). We used a publicly available scRNA-seq data set from bone marrow-derived HSPCs and analysed the expression of genes that encode cytosolic ribosomal proteins. After calculating average $\log_{10}$ expression profiles for each of the six different cell types (HSC, MPP, MLP, CMP, GMP and MEP), we calculated the pairwise Pearson correlation. The analysis revealed very strong correlations (0.92–0.99) between the ribosomal gene expression in HSCs and all five progenitor populations (Supplementary Fig. 9). To quantitatively assess whether the absolute expression value of each gene fluctuates, we calculated the coefficient of variation (CV)[36] for each ribosomal gene across the six different cell types. Our results suggest that for the case of cytosolic ribosomal genes, absolute gene expression values across different cell types showed low levels of fluctuation (CV score, 0.116 ± 0.047), whereas mitochondrial ribosomal genes were randomly expressed at different levels (CV score,

1.589 ± 1.033). This shows that ribosomal gene expression of human HSCs is highly similar to more mature progenitors, confirming an evolutionary conservation of this trend from zebrafish.

**HSPC transcriptome is conserved compared to mouse and human.** Zebrafish are an important model system in biomedical research and has been extensively used for the study of haematopoiesis. Although it has been demonstrated that many transcription factors and signalling molecules in haematopoiesis are well conserved between zebrafish and mammals, comparative analysis of the whole transcriptome was lacking.

In order to explore the evolution of blood cell-type-specific genes, we performed conservation analysis between zebrafish and other vertebrate species (see 'Methods' section). For this analysis, we enriched our initial data set with 81 natural killer (NK) and 109 T cells derived from the spleen of two adult zebrafish[37]. Our analysis revealed particularly high conservation of the HSPC transcriptome. For example, 90% of HSPC-specific genes in zebrafish had an ortholog in human and mouse compared to 70–80% of erythrocyte-, monocyte-, neutrophil- and thrombocyte-specific genes (Fig. 6a). The lowest conservation was observed for T cells (59%) and NK cells (68%), possibly reflecting their adaptation to fish-specific pathogens and virulence factors (Fig. 6a).

Gene duplication is the major process of gene divergence during the molecular evolution of species. We therefore analysed

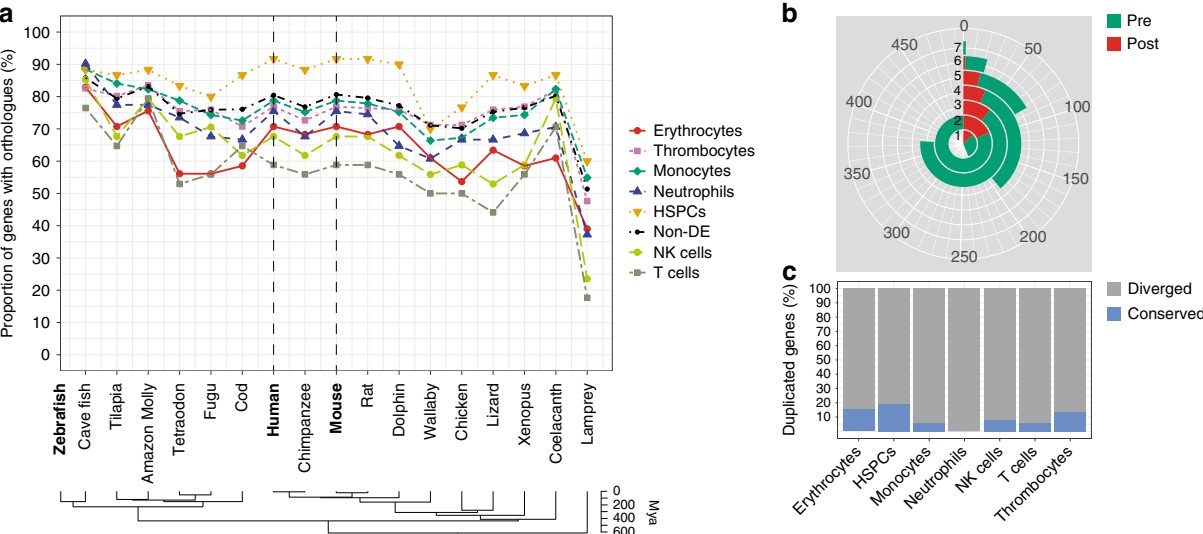

**Fig. 6** Conservation analysis of zebrafish genes differentially expressed in the main blood cell types. **a** Percentage of zebrafish protein-coding genes (specific for distinct blood cell types, as well as non-differentially expressed) with orthologs in other vertebrate species. **b** The total number of paralogs duplicated exclusively pre- (green) and post-ray-finned speciation (red). The numbers 1–7 mark the number of cell types (erythrocytes, monocytes, neutrophils, thrombocytes, HSPCs, T cells and NK cells) in which the duplicated genes are expressed. **c** The percentage of conserved vs diverged genes duplicated exclusively post speciation (fish-specific genes)

duplications that occurred exclusively before (referenced hereafter as pre-speciation genes) or after speciation (referenced hereafter as post-speciation genes) of the last common ancestor between fish (Actinopterygii) and mammals (Sarcopterygii)[37], (see 'Methods' section). Out of 7424 paralogs that were expressed in our data set (see 'Methods' section), around 79% were duplicated pre- and 21% were duplicated post speciation (Fig. 6b). Following ray-finned-specific duplication, the paralogs were more likely to functionally diverge (88%) and show expression in different cell types than to remain expressed in the same cell type (conserved expression), 12% (Fig. 6b, c). Interestingly, HSPCs had the highest percentage of paralogs (19%) with a conserved expression pattern (Fig. 6c). This number was lowest for duplicated genes in innate (0% for the neutrophils and 6% in monocytes) and adaptive immune cells (8% for the NK and 6% for the T cells). Altogether, our findings further underline the relevance of the zebrafish model system in advancing our understanding of the genetic regulation of haematopoiesis in both normal and pathological states.

**BASiCz**. The characterisation of mouse and human haematopoietic cells is dependent on the presence of cell surface markers and availability of antibodies specific for diverse progenitor populations. The antibodies for these cell surface markers are thus used to isolate relatively homogeneous cell populations by flow cytometry. Transcriptional profiling of isolated cell populations[38, 39] and more recently single cells[40] have further allowed genome-wide identification of cell-type-specific genes. However, beyond mouse and human, less is known about the transcriptome of blood cell types, mainly due to the lack of suitable antibodies.

To overcome this knowledge gap, we have generated a user-friendly cloud repository, BASiCz (Blood Atlas of Single Cells in zebrafish) for interactive exploration and visualisation of 31,953 zebrafish genes in 1422 haematopoietic cells across five different cell types. The generated database (http://www.sanger.ac.uk/science/tools/basicz) allows easy access and retrieval of sequencing data from zebrafish blood cells.

## Discussion

Cell differentiation during normal blood formation is considered to be an irreversible process with a clear directionality of progression from HSCs to more than 10 different blood cell types. It is, however, widely debated to what extent the process is gradual or direct[6, 13] on the cellular level; and in the case of the gradual model, what the intermediates of the increasingly restricted differentiation output of progenitor cells are[2–5, 33]. Although these models are very different in the way that they describe lineage progression, the identity of haematopoietic cells is determined based on the cell surface markers and the progression of cells during differentiation is defined on a cellular rather than transcriptional level.

Here we used a marker-free approach to order cells along their differentiation trajectory based on the transcriptional changes detected in the single-cell RNA-seq data set. Our analysis showed a gradual transition of cells on a global transcriptional level from multipotent to lineage restricted. The computationally reconstructed tree further revealed that differentiating cells moved along a single path in the 'state space'. This path included an early split of cells towards thrombocyte–erythrocyte and monocyte–neutrophil trajectories. However, cells in the 'middle' of the tree (HSPC state) showed considerable cell-to-cell variability in their probability to transition to any of the four cell types. This suggested that although global transcriptional changes before and after the branching point were continuous, the probability of a cell transitioning to any of the four committed states was determined only by a subset of highly relevant genes. Therefore, cells that were transcriptionally similar overall could have a high probability of differentiation to distinct cell types.

Interestingly, once the cell fate decision was executed, suppression of transcription of ribosomal genes and upregulation of genes which are relevant for the function of each cell type coordinately controlled lineage differentiation. Of all genes that were annotated as 'ribosomal proteins' on the Ensembl BioMart database, only those that encoded cytoplasmic ribosomal proteins showed dynamic expression in pseudotime in our data set. Importantly, this change was not linked to the expression of cell cycle-specific genes, excluding proliferation rates as a potential

reason for these data[35]. Furthermore, our analysis of data obtained from human HSCs and progenitors revealed that ribosomal gene expression levels are highly similar between the different progenitor types and stem cells[35].

Our comparative analysis between zebrafish and human across seven different haematopoietic cell types revealed a high overall conservation of blood cell-type-specific genes. Together with BASiCz, a user-friendly cloud repository, we generated a comprehensive atlas of single-cell gene expression in adult zebrafish blood. Data-driven classification of cell types provided high-resolution transcriptional maps of cellular states during differentiation. This allowed us to define the haematopoietic lineage branching map, for the first time, in zebrafish in vivo.

## Methods

**Zebrafish strains and maintenance**. The maintenance of wild-type (Tubingen Long Fin) and transgenic zebrafish lines[29–31, 41–45] (Supplementary Table 1) was performed in accordance with EU regulations on laboratory animals[46]. All experiments were approved by the Sanger Institute's Animal Welfare and Ethical Review Body.

**Single-cell sorting**. A single kidney from heterozygote transgenic or wild-type fish was dissected and placed in ice cold PBS/5% foetal bovine serum. At the same time, testes were dissected from the same fish. Single-cell suspensions were generated by first passing through a 40-µm strainer using the plunger of a 1 ml syringe as a pestle. These were then passed through a 20-µm strainer before adding 4′,6-diamidino-2-phenylindole (DAPI, Beckman Coulter, cat no B30437) for mCherry/dsRed2, or propidium iodide (PI, Sigma, cat no P4864) for GFP/EGFP. Individual cells were index sorted into wells of a 96-well plate using a BD Influx Index Sorter. Kidneys from a non-transgenic line were used as a control for gating[16].

**Whole transcriptome amplification**. The Smart-seq2 protocol[47, 48] was used for whole transcriptome amplification and library preparation[16]. In brief, single cells were lysed by incubation in a 0.2% Triton X-100 solution at 72 °C for 3 min. Next, cDNA was generated using SmartScribe enzyme (100 units per sample) and a template switching oligo (1 µM). At this step, we added 92 external RNA controls consortium (ERCC) spike-ins at a final dilution of 1:10. PCR pre-amplification was carried out using KAPA HiFi HotStart ReadyMix and 24 PCR cycles. The reaction product of the PCR was then purified using Ampure XP beads in conjunction with a magnetic stand, washing several times with 80% ethanol. At this step, cDNA quality was assessed using an Agilent Bioanalyzer and qPCR. Samples of sufficient quality were used for library preparation. For tagmentation, we used the Illumina Nextera XT DNA kit, incubating the mixture for 5 min at 55 °C. After stripping the transposase enzyme, adaptor ligation was carried out using the Nextera PCR master mix and index primers, cycling for 12 cycles. The library was then purified again using beads. Following a final Bioanalyzer quality check, the libraries were pooled and diluted to the concentration required by the different sequencers.

They were sequenced in paired-end mode on the Illumina Hi-Seq2500 or Hi-Seq4000 platforms.

**Cytology**. Sorted transgene-positive or gated wild-type cells were concentrated by cytocentrifugation at $7 \times g$ for 5 min onto SuperFrostPlus slides using a Shandon Cytospin 3 cytocentrifuge. Slides were fixed for 3 min in −20 °C methanol and stained with May–Grünwald Giemsa (Sigma). Images were captured using a Leica DM5000b microscope in conjunction with a ×63 oil-immersion lens and an Olympus DP72 camera.

**Transplantation experiments**. Adult $rag2^{E450fs−/−}$ mutant fish[32] were irradiated in an IBL 437 irradiator using a 10 Gy dose from a Caesium 137 source. After 1–2 days of recovery, donor cells were prepared from kidneys of transgenic fish as described above. Using the same gating strategy as employed for the single-cell sorting, fluorescent cells were collected by flow cytometry into microtubes containing 20 µl ice cold PBS/5% foetal bovine serum. Using a volume of 10 µl, 500 cells were transplanted into the anaesthetised (0.02% tricaine, Sigma A5040) $rag2^{E450fs−/−}$ recipients via intraperitoneal injection. As described above, engraftment into the whole kidney marrow was analysed by FACS at 1 day, 4 and 14 weeks post transplantation. The engrafted cells at 4 and 14 weeks post transplantation were single-cell index sorted and processed for single-cell RNA-seq as described above.

**Benchmarking single-cell RNA-sequencing methods**. One of the most important components that contributes to errors during the alignment and quantification of single-cell RNA-sequencing data is the presence of multi-mapped (or ambiguous) reads[49]. Currently, there are many different bioinformatic strategies that can

be used to align (e.g., STAR[50], Tophat[51], Bowtie[52], Salmon[53], Sailfish[54], etc.) and quantify scRNA-seq data (e.g., htseq[55], cufflinks[56], Salmon[53], Sailfish[54], etc.).

However, independent of the method applied, one of two possible strategies can be used to align reads, namely unique and multi-mapped. A comprehensive comparative analysis across many different scRNA-seq approaches has recently been published. It suggests that both setups (i.e., single and multi-mapped reads) are able to cope with ambiguous reads effectively[49].

In order to assess the impact of using a unique vs. multi-mapped reads alignment strategy on our data set, we re-analysed our raw data using STAR[50] in uniquely aligned reads mode. Salmon[53] was used next to quantify the transcripts. The Pearson correlation of the average gene expressions between Salmon and Sailfish at single-cell level ranged from 0.81 to 0.91, suggesting a strong correlation between alignments that included uniquely mapped reads and those that did not (Supplementary Fig. 1a). As expected, the number of detected genes (TPM > 1) was lower for Salmon compared to Sailfish (Supplementary Fig. 1b). However, the genes' variability distribution (CV) across single cells for each plate was comparable between the two methods (Supplementary Fig. 1c).

**Extended analysis of the reconstructed lineage tree in zebrafish**. To further investigate how robust our computational reconstruction of the lineage tree is, we applied different cutoffs to define variable genes. We next reconstructed the lineage tree using Monocle2[15]. Specifically, the highly variable genes were calculated using: 5% biological variation, 25% (default analysis) and 95% biological variation (three components). We then analysed the overall structure of the tree and the percentage of the misclassified cells as compared to the default setting that we used in the initial submission.

**Single-cell RNA-seq processing and QC**. Reads were aligned to the zebrafish reference genome (Ensemble BioMart version 83) combined with the EGFP, mCherry, tdTomato and ERCC spike-ins sequences. Quantification was performed using Sailfish[54] version 0.9.0 with the default parameters using paired-end mode (parameter –l IU).

Transcript per million (TPM) values reported by Sailfish were used for the QC of the samples. Wells with fewer than 1000 expressed genes (TPM > 1), or more than 60% of ERCC or mitochondrial content were initially annotated as poor quality cells (Supplementary Fig. 1). However, due to the lower number of expressed genes in erythroid cells, we further investigated the expression levels of adult globin genes, ba1 and hbaa1[26], in all erythroid cells. Based on comparison with the empty wells, samples that expressed both ba1 (>40,000 TPM) and hbaa1 (>9000 TPM) were considered to pass QC (Supplementary Fig. 2). Therefore, a total of 1422 single cells were selected for further analysis.

Average single-cell profiles compared to corresponding bulk wells revealed strong correlations (Pearson's correlation coefficient) ranging from 0.7 to 0.9 as illustrated in Supplementary Fig. 2, suggesting that the single-cell expression profiles were effectively quantified.

For each of the 1422 single cells, both gene and ERCC counts reported by Sailfish, were transformed into normalised counts per million (CPM). To do this, we divided the number of counts for each gene by the total number of counts (i.e., sum of all counts per cell) in each cell followed by multiplication of the resulting number by 1,000,000. The library size and cell-specific biases were removed (e.g., differences during amplification, ERCC concentration, batch effects, etc.) using the scran R package (version 1.3.0)[57]. Out of 31,953 genes, we retained those that were expressed in at least 1% of all cells (CPM > 1). Thus, a total of 20,960 genes were used for further analysis.

**Technical noise fit and identification of highly variable genes**. To distinguish biological variability from the technical noise in our single-cell experiments, we inferred the most highly variable genes using ERCCs as spike-in in all 1422 blood cells[36]. We used the scLVM[58] R package (version 0.99.2) to identify the 1845 most highly variable genes (Supplementary Fig. 3).

Principal component analysis (pcaMethods[59] (version 1.64.0)), independent component analysis (FastICA[60] (version 1.2) and diffusion maps (destiny[61] (version 1.3.4)) were used to verify that all cells were intermixed in the reconstructed 3D component space based on their transcriptional properties and not based on the fish or a plate they originated from.

**Pseudotime ordering DE and dynamically expressed genes**. The set of 1845 most highly variable genes was used to order the 1422 single cells along a trajectory using the Monocle2[15] R package (version 1.99.0). The 'tobit' expression family and 'DDRTree' reduction method were used with the default parameters. As illustrated in Fig. 1, cells ordered in the pseudotime created five distinct states. To assign identity to each of the five states, we performed differential expression (DE) analysis between each state vs. the remaining four using the 'differentialGeneTest' Monocle2 function. We modelled expression profiles of each state using a Tobit family generalised linear model[15]. For each state, statistically significant genes that scored $P < 0.01$, $q < 0.1$ (false discovery rate (FDR)) and were expressed in more than 50% of the cells were further used to perform GO analysis.

To enrich for HSPCs, we added 302 transplanted runx1+ cells to our previous data set for a total of 1724 cells. We re-analysed the data the same way as described

above and used the 1871 most variable genes for the calculation of a new Monocle trajectory.

Finally, we identified genes that change as a function of pseudotime across each of the four branches by setting the 'fullModelFormulaStr' parameter equal to '~sm.ns(Pseudotime)'. Genes whose expression changed dynamically in pseudotime were selected using the same statistical criteria as described for DE genes. For each branch, we clustered dynamically expressed genes using the 'plot_pseudotime_heatmap' function with the default parameters. The number of clusters (trends) in each branch was determined by its silhouette plot score (cluster R package version 2.0.5)[62]. To generate the trend lines across different states (see Fig. 3b), we used the average expression pattern of the dynamically expressed genes that follow the same trend across pseudotime and fit them using the ggplot2[63] R package (version 2.2.1) stat_smooth() parameter. We used the Gaussian linear model and the formula 'y ~ poly(x,2)' at 0.95 of standard error (grey area of the plot).

For the analysis of ribosomal genes, we used the Ensembl BioMart version 83 and selected all genes annotated with the term 'ribosomal protein'. We performed clustering using the pheatmap function (R pheatmap package version 1.0.8)[64] using Euclidean distance and ward.D2 linkage.

To investigate the correlation between ribosomal and cell cycle gene expression, we identified a total of 342 zebrafish genes annotated as 'GO:0007049', i.e., 'cell cycle' using BioMart (version 83). Next, we performed clustering between a subset of the cell cycle genes expressed in more than 10% of cells in each of the branches of the Monocle trajectory and dynamically expressed ribosomal genes using the tools described above.

**Analysis of human cells.** In order to show the generalisability of our findings from zebrafish to humans, we used a publicly available human single-cell RNA-seq data set[33], deposited in the Gene Expression Omnibus (GEO) under accession code GSE75478. This set contained data from 1344 single cells, which we aligned to the latest human reference genome (GRCh38p10 version 88) and quantified gene expression using Sailfish (version: 0.9.0). Following QC, we were left with 891 single cells, which included HSCs and various progenitor fractions (Supplementary Table 2). After normalisation with the scran package of the resulting CPMs (similarly to zebrafish data), we next identified 341 genes that were annotated as 'Ribosomal' using the BioMart database (GRCh38p10 version 88) and were expressed in more than 1% of all cells. Of these, 250 were expressed at a very low level in this data set. GO term enrichment analysis revealed that these genes encode mitochondrial ribosomes. In contrast, 91 genes that were expressed at a high level, encoded cytosolic ribosomal genes, as suggested by GO term enrichment analysis. Since our initial analysis using zebrafish cells focused only on genes that encode cytosolic ribosomes, we focused on the same population of genes in the human data set. Finally, we calculated the pairwise Pearson correlation between the cytosolic ribosomal genes for each progenitor population.

**GO analysis.** DE genes were ranked for each of the five states based on the mean $\log_{10}$ counts. Genes with average lower than 2 and those expressed in more than one state were not included in the GO analysis. GO analysis was performed using the gProfileR[65] package (version 0.6.1) using the gprofiler command with the following parameters: organism = 'drerio', hier_filtering = 'moderate', correction_method = 'fdr' and max_p_value = 0.05.

**Conservation analysis of the cell-type genes in zebrafish.** In order to perform the conservation analysis, we identified the orthologous genes (BioMart Ensembl version 83) between the zebrafish and other vertebrate species, including cave fish, tilapia, amazon molly, tetraodon, fugu, cod, human, chimpanzee, mouse, rat, dolphin, wallaby, chicken, lizard, Xenopus, coelacanth and lamprey. For this analysis, we enriched our initial data set with 81 NK and 109 T cells derived from the spleen of two adult zebrafish[37]. Following the same computational approach as we did with the initial data set, we re-calculated the DE genes for each of the seven different clusters. We only considered 'protein_coding' genes that were expressed in more than 50% of cells within each cluster and scored more than mean $\log_{10}$ counts. This resulted in 41 erythrocyte-, 113 monocyte-, 102 neutrophil-, 212 thrombocyte-, 60 HSPC-, 34 NK- and 34 T-specific genes that were used for further analysis. For the case of the non-DE genes, we included only 'protein_coding' annotated genes that were expressed in more than 1% of all cells (CPM > 1) and with average gene expression higher than the global mean of 0.10. The final list of the non-DE genes included 8127 genes.

**Analysis of duplicated genes in zebrafish.** In order to analyse duplicated genes[37], we first identified all zebrafish 'protein_coding' paralog genes listed in Ensembl (BioMart Ensembl version 83) and split them into two groups: (1) 17,158 pre-ray-finned fish duplicated genes, including Euteleostomi, Bilateria, Chordata, Vertebrata and Opisthokonta parent taxa, and (2) 11,806 post-ray-finned fish duplicated genes, including Neopterygii, Otophysa, Clupeocephala and Danio rerio children taxa. We next removed duplicated genes that were found in common between the two groups. This resulted in 8601 pre-, and 3249 post-ray-finned fish genes that we used in further analysis.

For the analysis of the expression pattern divergence, we focused on genes that were expressed in our data set. We analysed expression pattern of all paralogs of DE genes (i.e., erythrocytes, monocytes, neutrophils, thrombocytes, HSPCs, NK- and T cells) that were expressed in more than 10% of cell in each of the branches (cell states). The expression pattern was considered to be conserved if duplicated genes and their annotated paralogs were all expressed in the same cell type. However, if at least one of the paralogs was expressed in a different cell type, this was considered as an example of potential functional divergence.

**Deep neural network DNN classifier.** To generate the deep neural network (DNN) model, we used Keras[66], a Python-based deep learning library for Theano[67] and Tensorflow[68]. We worked with the Keras functional API, which allows the definition of complex systems, such as multi-output models.

The DNN was used to predict the probabilities of a specific gene expression profile to be classified into one of the four differentiated cell types. We used the entire set of genes for all differentiated cells in the branches (1724 cells in total), i.e., erythrocytes, thrombocytes, neutrophils and monocytes. The input was therefore formed by 20,960 nodes (genes), which were normalised using z-values or standard scores. For the hyper-parametric fine-tuning of the DNN, we generated and evaluated models with different number of hidden layers, hidden nodes, network initialisations, regularisations and batch normalisation. The final hyper parameters were chosen according to the optimal performance and convergence of the accuracy and loss values.

The model was comprised of two hidden layers with 100 and 50 nodes, using a weight decay regularisation with a $\lambda$-value of 0.001, and Gaussian dropout of 0.8 between them. The chosen activation functions were 'relu' for the hidden layers, and 'softmax' for the output. The validation was performed over 20% of the initial data set, using 'categorical cross-entropy' loss. The average classification accuracy after convergence was $0.998 \pm 0.002$, and cross entropy loss of $0.03 \pm 0.004$, validation accuracy of $0.964 \pm 0.003$ and cross entropy validation loss $0.15 \pm 0.008$.

The neural network output returns the probability of a gene expression input vector (cell) to be classified as each one of the differentiated cell types. We can use these probabilities and their distributions to generate a value that determines the 'Stemness' of the cells according to the NN output. The 'Stemness value' is a measure of similarity between the input vector and the average distributions for each output class, which can be then used to indicate the cell differentiation state of the input.

This measure has been previously[33] used for similar purposes. It is based on the Kullback–Leibler distance between probabilities, and the 'Stemness value' ($S_i$) of cell $i$ is determined by the equation:

$$S_i = \sum_{j=1}^{N_c} p_{ij} \log \frac{p_{ij}}{\overline{p}_j}$$

Where $N_c$ is the number of classes, $\overline{p}_j$ is the average probability of class $j$ and $p_{ij}$ is the probability of cell $i$ to belong to class $j$.

**Cloud repository.** We have generated a cloud repository to enable research community to access single-cell gene expression profiles of 1422 zebrafish blood cells across all the 31,953 zebrafish genes. The implementation of the cloud service was performed using shiny[69] (version 0.14.2), https://shiny.rstudio.com, and plotly[70] (version 4.5.6), https://plot.ly, R packages.

**Statistics and reproducibility of experiments.** Statistical tests were carried out using R software packages as indicated in the figure legends and in the 'Methods' section. No statistical method was used to predetermine sample sizes. No randomisation or blinding of samples was performed. Pearson correlation coefficient was used to compare the average profiles of single cells against the bulk. Significance of differentially expressed genes was calculated with an approximate likelihood ratio test (Monocle2 differentialGeneTest() function) of the full model '~state' cells against the reduced model '~1'. For the dynamically expressed genes, the full model '~sm.ns(Pseudotime)' was tested against the reduced model of no pseudotime dependence. In both cases, P values were normalised using the the Benjamini–Hochberg FDR, selecting statistically significant genes with P < 0.01 and FDR < 0.1. For the GO analysis, the hypergeometric test (equivalent to the one tailed Fisher's exact test) was used to evaluate the significant terms, while P values were corrected for multiple testing using the FDR approach, with FDR < 0.05 considered statistically significant, using the gProfiler R[65] package.

**Data availability.** Raw data can be found under the accession number E-MTAB-5530 on ArrayExpress. Additional Zebrafish-related RNA-seq data that were used in the present study can be found in E-MTAB-4617, E-MTAB-3947 while human-related data were collected from the GEO under accession code GSE75478.

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

## Acknowledgements

The study was supported by Cancer Research UK grant number C45041/A14953 (to A.C. and E.A.), European Research Council project 677501—ZF_Blood (to A.C.) and a core support grant from the Wellcome Trust and MRC to the Wellcome Trust − Medical Research Council Cambridge Stem Cell Institute. The authors thank WTSI Cytometry Core Facility for their help with index cell sorting and the Core Sanger Web Team for hosting the cloud web application. The authors also like to thank the CRUK Cambridge Institute Genomics Core Facility for their contribution in sequencing the data.

## Author contributions

E.I.A. carried out the analysis; J.G.B. and L.F. performed the experiments; H.A. generated the DNN; P.L. oversaw implementation of the DNN; J.G.B., E.I.A. and A.C. contributed to the discussion of the results and designed the figures; A.C. conceived the study and wrote the manuscript. All authors approved the final version of the manuscript.

## Additional information

**Competing interests:** The authors declare no competing financial interests.

