## [Peer Review File · Nature Communications]

Reviewers' comments:

Reviewer #1 (Remarks to the Author):

In this manuscript, Athanasiadis et al analyzed the transcriptional state of hematopoietic cell types in zebrafish by single-cell RNA-seq. Using a group of genes (1,845) identified as the most highly variable genes, they computationally reconstructed the blood lineage tree. They identified 5 distinct cell 'states', which were named as monocyte, neutrophils, HSPCs, erythrocytes, and thrombocytes, based on the gene expression patterns and associated GO terms. By comparing transcriptional differences within each cell state and at the branching point, they identified interesting patterns of gene expression changes including the downregulation of ribosomal genes. They further compared the zebrafish transcriptome to other vertebrate species including mouse and human to correlate transcriptomic changes vs cell fates.

Overall, this study provides a nice resource for single cell RNA-seq-based transcriptomic analysis of zebrafish blood cells. In-depth analysis of the transcriptional states provides new insights into the transitional stages and associated gene expression changes in lineage specification. Transplantation analysis using transgenic models provide important validation of some of the defined cell states. Several interesting patterns of transcriptomic changes, such as the downregulation of ribosomal genes during specification, were described although the functional relevance of these changes remains unexplored. Moreover, the 'resolution' of the defined 'HSPC state' based on current analysis seems to be low and it remains unclear how heterogeneous it is, and whether or not distinct transitioning states can be further identified.

The specific comments:

1, Using 1,845 most highly variable genes, the authors defined 5 distinct cell 'states', with 4 correspond to known mature lineages (monocyte, neutrophils, erythrocytes and thrombocytes) and one as 'HSPC state'. How stable are the computationally reconstructed states if using different sets of genes, e.g. variable genes defined by different cutoffs or all genes detected by single-cell RNA-seq? Can the resolution be further improved for HSPC to provide more information about the heterogeneity (see below comment #3) and the refined transitioning states? To address these questions, it may be helpful to perform similar analyses under certain perturbative conditions (e.g. knockout or knockdown of known factors required for HSPC or other lineages) and compare the results.

2, The findings related to the downregulation of ribosomal genes in pseudotime are intriguing, but the functional relevance remains unexplored. The overall impact of this study would be greatly enhanced if the authors could provide additional functional and/or mechanistic evidence supporting the role of the identified pathways in cell state transitioning during hematopoiesis.

3, The authored discussed that the findings related to expression changes in ribosomal genes are not in line with previous studies showing lower rates of protein synthesis in HSCs (Signer et al., 2014 Nature). It is important to know that, in the Signer et al study, protein synthesis rate in prospectively isolated distinct hematopoietic populations including HSC, various progenitors (MPP, CMP, GMP, and MEP), and mature lineages were compared. They found that the protein synthesis rate was the lowest in HSCs, but significantly upregulated in various progenitor populations (CMP, GMP and MEP). Given the relatively 'low resolution' of the HSPC state (as described in #1 above), it remains unclear whether the discrepancy is due to the heterogeneity of the HSPC state defined in this study. Therefore, it will be important to further analyze the changes in ribosomal genes (and other identified patterns of gene expression) if the resolution of HSPC can be further enhanced.

4, A minor point, how was the normalization of gene expression against total level of transcripts performed? It is important to know whether any changes in the total transcript levels at the single

cell level and how they affect the identification of variable genes used to define the cell states. The reviewer can't seem to find such information in the text or methods.

Reviewer #2 (Remarks to the Author):

In this study, Athanasiadis et al use single cell RNA-sequencing to transcriptionally define zebrafish hematopoiesis. Computational construction of the blood lineage tree suggests that lineage commitment of hematopoietic stem and progenitor cells occurs through a single path and intermediate progenitor. The authors further demonstrate that lineage commitment is associated with transcriptional repression of ribosomal genes. In some aspects, this study is consistent with several other transcriptional profiling and functional studies that have defined the presence of unipotent myeloid and erythroid progenitors. Unfortunately, this study seems significantly underpowered to faithfully reconstruct the complexity of native hematopoiesis.

1. The central branch of the reconstructed hematopoietic tree contains hematopoietic stem and progenitor cells. Despite the transcriptional clustering, this is a functionally heterogeneous population of cells, and likely contains very few true HSCs. Transplantable HSCs are present in the zebrafish kidney at a frequency of $\sim 1/38,000$ (Hess et al., PNAS 2013; 110:4327). To what extent are HSCs enriched by the use of transgenic zebrafish in this study? How many of 1,422 sequenced cells are likely to be true HSCs?

2. The likely absence of significant numbers of HSCs complicates the interpretation of the ribosomal gene expression data. Studies demonstrating low rates of protein synthesis within HSCs were performed using highly purified cell populations (based on functionally validated cell surface marker profiles). These same studies demonstrated that many progenitors have very high rates of protein synthesis. Since the HSPC branch in the present study does not distinguish between HSCs and progenitors, and HSCs are likely to be an exceedingly rare contributor to this population, the relatively high expression of ribosomal genes is most simply explained as a consequence of the high progenitor content of this population.

3. The observation that ribosome gene expression decreases as cells differentiate is not unexpected. Ribosome biogenesis is a key feature of cell proliferation, and rapidly dividing progenitors would depend upon ribosome production more than terminally differentiated cells with limited (or no) proliferative potential. How does ribosome gene expression correlate with cell cycle gene expression?

4. Why are there no lymphoid lineage cells present in this reconstruction? Were they deliberately excluded?

Reviewer #3 (Remarks to the Author):

This manuscript presents the transcriptional profiles of 1,422 zebrafish haematopoietic cells by single-cell RNA sequencing (scRNA-seq) technology. The authors performed quite a few scRNA-seq analyses, including quality control, gene quantification, cell subtyping, differentiation trajectory mapping, differential expression analysis between different defined cell types. Then the authors reconstructed the blood lineage tree, defined 5 distinct cell "states", investigated the differential expression patterns between each state and identified a pool of state-specific expression marker genes. In addition, they also compared the expressions of ribosomal genes and state-specific genes during lineage differentiation, analyzed the conservation of those state-specific genes between zebrafish and mammals, and built a cloud repository, BASiCz, for data visualization. Overall, this study provides a fantastic resource for dissecting haematopoiesis at the

transcriptome level.

In this study, the authors performed 1,422 scRNA-seq analyses for zebrafish haematopoietic cells, comprehensively investigated the differentiation trajectories and state-specific expression patterns, built a cloud repository for the data visualization, and finally analyzed the conservation of haematopoietic program between zebrafish and mammals. Considering that this manuscript is well-organized and clearly-written, I only give two minor comments:

1. In the scRNA-seq data processing, I noted that the authors used Sailfish (without alignment) tools to quantify gene expression. However, uniquely-aligned reads are commonly recommended for scRNA-seq data processing (Grun, D. and A. van Oudenaarden (2015). Cell 163(4): 799-810.). Have authors compared two different processing strategies ("alignment + quantification with uniquely-mapped reads" vs. "quantification without alignment")? How does it affect the accuracy of gene quantification?
2. Please cite the papers of software/tools used in this study if applicable, such as the pcaMethods R package used in this study.

We would like to thank all three Reviewers for carefully reading our manuscript and for their highly valuable comments. We have addressed the issues and concerns that were raised. Furthermore, the study has been expanded by including sequencing data from about 300 new *runx1+* single cells from transplantation recipients to increase the number of HSCs in our dataset. First, we summarize the additional data and analyses that we performed. Next, we address the Reviewers' responses in the same order as in their respective reports. Our responses are in italics and quotes from the revised manuscript are in blue.

While all new data are included in the rebuttal letter, not all are included in the revised manuscript. We would therefore appreciate further input from Reviewers and editor on whether they would like us to incorporate the whole new data set in the revised manuscript.

Summary of the additional data and analyses

1) Extended analysis of the reconstructed lineage tree in zebrafish

To further investigate how robust our computational reconstruction of the lineage tree is, we applied different cutoffs to define variable genes. We next reconstructed the lineage tree using Monocle2. Specifically, the highly variable genes were calculated using: 5% biological variation, 25%- (default analysis) and 95% biological variation (three components). We then analysed the overall structure of the tree and the percentage of the misclassified cells as compared to the default setting that we used in the initial submission (Fig. 1).

Figure 1: Monocle trajectories generated using different sets of highly variable genes. Trajectories were generated using three different sets of highly variable genes. Highly variable genes were calculated using thresholds of 5%, 25%-(default) and 95% biological variation. For each reconstructed tree, the percentage (and a total number) of “misclassified” cells in each branch was calculated in comparison to the default setup (25%). k =number of components used.

Our analysis shows that the overall structure of the lineage tree is quite robust even when the cutoff is significantly changed. The main difference between the three models is that a small number of cells in the HSPC cluster, close to the branching point may move into the nearest branch. This is not entirely surprising, as none of the currently available methods is able to determine the exact branching point and some variations are therefore possible. We were able to largely resolve this issue with the use of machine learning approaches (for details see the point 3 in this summary).

2) Benchmarking single-cell RNA sequencing methods

One of the most important components that contributes to errors during the alignment and quantification of single-cell RNA-Sequencing data is the presence of multi-mapped (or ambiguous) reads (Robert and Watson, 2015). Currently, there are many different bioinformatic strategies that can be used to align (e.g. STAR, Tophat, Bowtie, Salmon, Sailfish, Kallisto etc) and quantify scRNA-seq data (e.g. htseq, cufflinks, Salmon, Sailfish, Kallisto).

However, independent of the method applied, one of two possible strategies can be used to align reads, namely unique and multi-mapped. A comprehensive comparative analysis across many different scRNA-seq approaches has recently been published. It suggests that both setups (i.e. single and multi-mapped reads) are able to cope with ambiguous reads effectively (Robert and Watson, 2015).

In order to assess the impact of using a unique versus multi-mapped reads alignment strategy on our data set, we re-analysed our raw data using STAR (Dobin et al., 2013) in uniquely aligned reads mode. Salmon (Patro et al., 2017) was used next to quantify the transcripts. The Pearson correlation of the average gene expression between Salmon and Sailfish at single cell level ranged from 0.81 to 0.91, suggesting a strong correlation between alignments that included uniquely-mapped reads and those that did not (Fig. 2). As expected, the number of detected genes (TPM > 1) was lower for Salmon compared to Sailfish (Fig. 3). However, the genes' variability distribution (Coefficient of Variation CV) across single cells for each plate was comparable between the two methods (Fig. 4). Please see page 19 onwards for details.

Figure 2. Pearson correlation of the average gene expressions between Salmon and Sailfish. Gene quantification accuracy was assessed by selecting the average $\log_{10}(TPM+1)$ gene expression for each of the 21 sequenced plates.

Figure 3. Violin plots of the number of detected genes ($TPM>1$) at single-cell level. Salmon (unique mapped reads) and Sailfish (multi-mapped reads) were compared for each of the 21 plates.

Figure 4. Distribution of the Coefficient of Variation. Comparison of $\log_{10}(CV+0.001)$ gene expression ($TPMs$) values at a single-cell level between Salmon (unique mapped reads) and Sailfish (multi-mapped reads), across 21 plates.

3) Sequencing additional cells and extending computational analysis of the heterogeneity of the HSPC population

Our computationally reconstructed lineage tree identified four main branches, as well as a “middle” part, which we defined as HSPCs. As pointed out by two Reviewers, the HSPC population contained a mixed population of a few true HSCs and many progenitor cells. To further increase the number of HSCs in our data set, we carried out single-cell RNA-Seq of **153 new runx1+** single cells (two 96-well plates) from additional transplanted fish. Furthermore, we included 149 runx1+ single cells from a previous transplantation experiment in the analysis. Taken together, we analysed an additional **302 runx1+** cells in the revised manuscript. We re-analysed the whole data set (1,724 cells in total), calculated most highly variable genes ($n=1,871$) and generated a new Monocle trajectory (Fig. 5).

Figure 5: Updated Monocle trajectory. We have reconstructed the Monocle trajectory using 1,724 single cells and 1,871 highly variable genes, following the same computational strategy as described in the initial analysis.

Next, we computed the stemness S^{rel} index to predict the potential HSCs in our data set using the Kullback–Leibler distance of the predicted probabilities compared to the expected one, for each of the four different branches (Velten et al., 2017) (Fig. 6a). The lower the “stemness” factor, the higher the confidence that a particular cell is a stem cell. Using a threshold of 3 sigma over the mean stemness value (0.05), our analysis predicted that 35 out of 214 cells in the middle part of the tree are HSCs. The majority of cells that were identified as stem cells originated from the *cd41* (13 cells) and *runx1* (14 cells) transgenic lines. It should be noted that both these lines have been previously identified to contain transplantable HSC (Tamplin et al., 2015) lending further confidence in our computational prediction.

Figure 6. Transcriptionally similar cells display different probabilities of being stem cells. a) Cells predicted to be stem cells in the middle part of the lineage tree according to their stemness index. The insert shows the new Monocle tree including transplanted cells (1,724 single cells and 1,871 highly variable genes). b) Distribution of stemness scores in different branches of the tree showing the presence of potential HSCs exclusively in the HSPC branch. c) Contribution of different transgenic lines to predicted stem cells.

4) Single cell analysis of a ribosomal gene expression in human HSCs and progenitor cells

We used a publicly available human single-cell RNA-Seq data set (deposited in the Gene Expression Omnibus (GEO) under accession code GSE75478 and published in (Velten et al., 2017). We downloaded raw data derived from 14 different 96-well plates (~215 GB raw data, 1,344 human single cells in total) and aligned it to the latest human reference genome (GRCh38p10 version 88). We quantified gene expression using Sailfish (version: 0.9.0). Both alignment and quantification, as well as all computational calculations were performed on the Sanger's traditional high performance compute (HPC) clusters (approximately 20,000 cores of computational resource, using IBM Spectrum LSF DRM, 13.5Pb of lustre clustered file systems, 160Gb/s network backbone). We then analysed 891 single cells that included all phenotypic HSCs and progenitors:

- 109 HSC (CD34+CD38-CD45RA-CD90+CD49f+)
- 179 MPP (CD34+CD38-CD45RA-CD90-)
- 64 CMP (CD34+CD38+CD10-CD45RA-CD135+)
- 247 GMP (CD34+CD38+CD10-CD45RA+CD135+)
- 224 MEP (CD34+CD38+CD10-CD45RA-CD135-)
- 68 MLP (CD34+CD38-CD45RA+).

We next identified 341 genes that were annotated as “Ribosomal” using the BioMart database (GRCh38p10 version 88) and were expressed in more than 1% of all cells. Of these, 250 were expressed at a very low level (average gene expression levels < 4 CPM) in this data set. GO term enrichment analysis revealed that these genes encode mitochondrial ribosomes. In contrast, 91

genes that were expressed at a high level (average gene expression levels > 400 CPM) encoded cytosolic ribosomal genes, as suggested by GO term enrichment analysis. Since our initial analysis using zebrafish cells focused only on genes that encode cytosolic ribosomes, we selected the same population of genes in the human data set. The heatmap showing expression of ribosomal genes in human HSCs and progenitor cells is illustrated below (Fig. 7).

Figure 7: Heatmap of ribosomal gene expression in human HSCs and progenitors. Clustering of all human ribosomal expressed genes across the different HSPCs, using Euclidean distance and Ward Linkage.

After calculating average \log_{10} expression profiles for each of the six different cell types, namely HSC, MPP, MLP, CMP, GMP and MEP, we calculated the Pearson correlation pairwise. The analysis revealed very strong correlations (0.92-0.99) between the HSCs and all five progenitor populations (Fig. 8). Therefore, even though HSCs have 10 fold higher *de novo* protein synthesis (Signer et al., 2014) their level of expression of genes that encode ribosomal proteins is similar (highly correlated) to that of the progenitor populations.

Figure 8: Correlation analysis of expressed ribosomal genes across 891 HSPCs. Pairwise Pearson correlation revealed similar expression levels of the expressed cytosolic ribosomal genes in HSC and various progenitors (MPP, CMP, GMP, MEP and MLP).

Responses to the Reviewers' comments

Reviewer #1 (Remarks to the Author):

In this manuscript, Athanasiadis et al analyzed the transcriptional state of hematopoietic cell types in zebrafish by single-cell RNA-seq. Using a group of genes (1,845) identified as the most highly variable genes, they computationally reconstructed the blood lineage tree. They identified 5 distinct cell 'states', which were named as monocyte, neutrophils, HSPCs, erythrocytes, and thrombocytes, based on the gene expression patterns and associated GO terms. By comparing transcriptional differences within each cell state and at the branching point, they identified interesting patterns of gene expression changes including the downregulation of ribosomal genes. They further compared the zebrafish transcriptome to other vertebrate species including mouse and human to correlate transcriptomic changes vs cell fates.

Overall, this study provides a nice resource for single cell RNA-seq-based transcriptomic analysis of zebrafish blood cells. In-depth analysis of the transcriptional states provides new insights into the transitional stages and associated gene expression changes in lineage specification. Transplantation analysis using transgenic models provide important validation of some of the defined cell states. Several interesting patterns of transcriptomic changes, such as the downregulation of ribosomal genes during specification, were described although the functional relevance of these changes remains unexplored. Moreover, the 'resolution' of the defined 'HSPC state' based on current analysis seems to be low and it remains unclear how heterogeneous it is, and whether or not distinct transitioning states can be further identified.

The specific comments:

1, Using 1,845 most highly variable genes, the authors defined 5 distinct cell ‘states’, with 4 correspond to known mature lineages (monocyte, neutrophils, erythrocytes and thrombocytes) and one as ‘HSPC state’. How stable are the computationally reconstructed states if using different sets of genes, e.g. variable genes defined by different cutoffs or all genes detected by single-cell RNA-seq? Can the resolution be further improved for HSPC to provide more information about the heterogeneity (see below comment #3) and the refined transitioning states? To address these questions, it may be helpful to perform similar analyses under certain perturbative conditions (e.g. knockout or knockdown of known factors required for HSPC or other lineages) and compare the results.

> To further investigate how robust our computational reconstruction of the lineage tree is, we applied different cutoffs to define variable genes. We next reconstructed the lineage tree using Monocle2. Specifically, the highly variable genes were calculated using: 5% biological variation, 25%- (default analysis) and 95% biological variation (three components). We then analysed the overall structure of the tree and the percentage of the misclassified cells compared to the default setting that we used in the initial submission. Our analysis suggests that the overall structure of the lineage tree is quite robust. The misclassified cells were mostly from the middle part of the tree, close to the branching point, which “moved” to one of the four branches. This is not entirely surprising as none of the currently available methods is able to determine the exact branching point, making some variation inevitable. This issue could be resolved to a good extent with the use of machine learning approaches.

Figure 1: Monocle trajectories generated using different sets of highly variable genes. Trajectories were generated using three different sets of highly variable genes. Highly variable

genes were calculated using thresholds of -5%, 25%(default) and 95% biological variation. For each reconstructed tree, the percentage (and a total number of cells) of “misclassified” cells in each branch was calculated compared to the default setup (25%). The colors from our original analysis/clustering were superimposed on the new trajectories to assess the number of misclassified cells.

>We also used all genes that were expressed in more than 1% of all cells (i.e. 20,960 genes) to reconstruct the differentiation trajectories. Independent on the number of components (k) we used (2, 3, 4, 5, 6 and 10) the generated trajectories were distorted (2 and 3) or pseudotime inference was not possible (4, 5, 6 and 10) even if cell-type specific populations clustered together (Fig. 9). However, we should point out that Monocle was not designed to efficiently handle large data sets in full scale. It is stated in Monocle’s documentation that use of all genes may “confuse” the algorithm due to the noise and that the algorithm might “fix” on a feature of the data that isn’t biologically meaningful (<http://www.bioconductor.org/packages/release/bioc/vignettes/monocle/inst/doc/monocle-vignette.pdf> please see Section 4.2.1 - Choosing genes for ordering). Therefore, Monocle should not be used on all detected genes. Once highly variable genes are selected the algorithm is quite robust. We have not included this analysis in the revised manuscript but would be happy to do that if the Reviewer deems it necessary to be included.

Figure 9: Inferring single cell trajectories using all genes. We calculated the Monocle trajectories using all genes with components k equal to 2, 3, 4, 5, 6 and 10. For the visualisation of the trajectories with components k equal 4 or higher, t-distributed stochastic neighbor embedding (tSNE R package version 0.1-3) transformation was implemented to transform the k dimensions into 2. The colors from our original analysis/clustering were superimposed on the new data.

> The question related to the resolution of the middle part of the tree is very interesting and we have decided to apply a new computational approach to resolve it. Specifically, we identified potential stem cells by calculating their stemness index, as described on page 4 of this document. We found that there were 35 potential stem cells in the middle part of our haematopoietic lineage tree.

> *Further experimental approaches are necessary to validate these predictions and this is something we are planning to do as a continuation of this work. This would include the generation of new transgenic lines, based on our computational predictions, to label HSCs in zebrafish. We believe this would be a more appropriate approach to identify true HSCs as compared to using knock down of known TFs relevant for HSCs. Zebrafish mutant lines for *cmyb* and *runx1* have been generated, but in these lines haematopoiesis is considerably perturbed and often limited to larval haematopoiesis (due to the poor survival of mutants). Therefore, we believe, single cell analysis of mutants would not be easily comparable with the wt haematopoiesis.*

2, The findings related to the downregulation of ribosomal genes in pseudotime are intriguing, but the functional relevance remains unexplored. The overall impact of this study would be greatly enhanced if the authors could provide additional functional and/or mechanistic evidence supporting the role of the identified pathways in cell state transitioning during hematopoiesis.

> *We agree with the reviewer that this finding is intriguing. However, understanding the relevance and mechanisms involved would not be feasible within a reasonable time frame as part of this revision. Although it has been nicely demonstrated that HSCs have lower rates of protein synthesis compared to various progenitors, our analysis suggests that this change is not correlated with the transcription of ribosomal genes (for details please see below). In other words, similar to the middle part of our tree, which shows high levels of ribosomal transcripts, HSCs and progenitor cells in human have similar, highly correlated levels of ribosomal transcripts. Understanding the mechanisms of this discrepancy, although very interesting is, we believe, outside the scope of this manuscript.*

3, The authored discussed that the findings related to expression changes in ribosomal genes are not in line with previous studies showing lower rates of protein synthesis in HSCs (Signer et al., 2014 Nature). It is important to know that, in the Signer et al study, protein synthesis rate in prospectively isolated distinct hematopoietic populations including HSC, various progenitors (MPP, CMP, GMP, and MEP), and mature lineages were compared. They found that the protein synthesis rate was the lowest in HSCs, but significantly upregulated in various progenitor populations (CMP, GMP and MEP). Given the relatively 'low resolution' of the HSPC state (as described in #1 above), it remains unclear whether the discrepancy is due to the heterogeneity of the HSPC state defined in this study. Therefore, it will be important to further analyze the changes in ribosomal genes (and other identified patterns of gene expression) if the resolution of HSPC can be further enhanced.

> *We completely agree with the reviewer and the concerns raised are appropriate and well justified. To address this issue we decided to take two independent approaches:*

*First, we included an additional 302 *runx1*+ cells in the analysis (of which 153 cells were newly sequenced) to increase the total number of HSCs in our data set. We then applied computational approaches to predict the potential HSCs and thus increase the resolution of our HSPC cluster.*

Second, we re-analysed publicly available human single-cell data set from well-defined HSC and progenitor populations and tested the level of transcription of genes that encode ribosomal proteins between HSCs and five different progenitor populations.

> *1. We re-analysed our new data set (1,724 cells in total) and generated a new Monocle trajectory (Fig. 5). Next, to predict the potential HSCs in our data set, we computed the S^{rel} index, using the Kullback–Leibler distance of the predicted probabilities compared to the expected one,*

for each of the four different branches (Velten et al. 2017). The lower the “stemness” factor, the higher probability that the cell was a stem cell. Using the threshold of 0.05, our analysis predicted that 35 out of 214 cells in the middle part of the tree are HSCs. Finally, we compared the expression of the genes that encode ribosomal proteins between predicted HSC and progenitors in the middle part of the reconstructed tree. Our analysis revealed that there is a strong correlation (0.989) of the average expression levels of the cytosolic ribosomal genes between the 35 predicted HSC and the 214 cells of the HSPC part of the revised tree, suggesting that all cells within the HSPC state express the specific set of ribosomal genes at very similar levels.

Modified text (page 6):

“To increase the number of HSCs in our data set and the resolution in the HSPC branch of the Monocle trajectory, we added the 302 transplanted *runx1* cells to our 1,422 previously sequenced cells. We re-analysed the whole data set (1,724 cells in total), and generated a new Monocle trajectory (Fig. 4a).

Next, we considered the frequency of potential HSCs in this data set. To do so, we computed the stemness S^{rel} index (Velten et al., 2017), using the Kullback–Leibler distance of the predicted probabilities compared to the expected one, for each of the four different branches (Fig. 4a and b). The lower the “stemness” factor, the higher the confidence that a particular cell is a stem cell. Using the threshold of 3 sigma over the mean stemness value (0.05), our analysis predicted that 35 out of 214 cells in the middle part of the tree are potential HSCs. The majority of cells that were identified as stem cells originated from the *cd41* (13 cells) and *runx1* (14 cells) transgenic lines (Fig. 4c). It should be noted that both these lines have been previously identified to contain transplantable HSC (Tamplin et al., 2015, Ma 2012) lending further confidence in our computational prediction. This suggests that, although both stem and progenitor cells are intermixed on the trajectory due to their overall similar transcriptomes, their lineage potentials (and thus stemness scores) are distinct.”

>2. We re-analysed a publicly available human single-cell RNA-Seq data set (deposited in the Gene Expression Omnibus (GEO) under accession code GSE75478 and published (Velten et al., 2017). We downloaded raw data and analysed 891 cells in total from this study - 109 HSC (CD34+ CD38- CD45RA- CD90+ CD49f+), 179 MPP (CD34+ CD38- CD45RA- CD90-), 64 CMP (CD34+ CD38+ CD10- CD45RA- CD135+), 247 GMP (CD34+ CD38+ CD10- CD45RA+ CD135+), 224 MEP (CD34+ CD38+ CD10- CD45RA- CD135-) and 68 MLP (CD34+ CD38- CD45RA+).

We next identified 341 genes that were annotated as “Ribosomal” using the BioMart database (GRCh38p10 version 88) and were expressed in more than 1% of all cells. Of these, 250 were expressed at a very low level in this data set. GO term enrichment analysis revealed that these genes encode mitochondrial ribosomes. In contrast, 91 genes that were expressed at a high level encoded cytosolic ribosomal genes, as suggested by GO term enrichment analysis. Since in our initial analysis in zebrafish we focused only on genes that encode cytosolic ribosomes, we selected the same population of genes in the human data set. The heatmap showing expression of ribosomal genes in human HSCs and progenitor cells is illustrated below (Fig. 7).

After calculating average log10 expression profiles for each of the six different cell types, namely HSC, MPP, MLP, CMP, GMP and MEP, we calculated the Pearson correlation pairwise. The analysis revealed very strong correlations (0.92-0.99) between the HSCs and all five progenitor populations (Fig. 8). Therefore, even though HSCs have 10-fold higher de novo protein synthesis

(Signer et al., 2014) their level of expression of genes that encode ribosomal proteins is similar (highly correlated) to that of the progenitor populations.

Modified text (pages 7 and 10):

“A previous study showed that the rate of protein synthesis in murine HSCs is considerably lower than that of progenitor populations (i.e. CMPs, GMPs and MEPs) (Signer et al 2014). This is not in line with our transcriptional analysis, which showed a decrease in ribosomal gene expression during differentiation. In order to address this discrepancy, we considered the correlation in ribosomal gene expression between human phenotypic HSCs (CD34+CD38-CD45RA-CD90+CD49f+) and the different progenitor fractions (for details please see Methods). We used a publicly available scRNA-Seq data set from bone marrow derived HSPCs and analysed the expression of genes that encode cytosolic ribosomal proteins. After calculating average log10 expression profiles for each of the six different cell types (HSC, MPP, MLP, CMP, GMP and MEP), we calculated the pairwise Pearson correlation. The analysis revealed very strong correlations (0.92-0.99) between the the ribosomal gene expression in HSCs and all five progenitor populations (Supplementary Fig. 9). Therefore, even though HSCs have 10-fold higher de novo protein synthesis (Signer et al., 2014) their level of expression of genes that encode ribosomal proteins is similar (highly correlated) to that of the progenitor populations. Our results described above suggested that there is a poor correlation between the level of transcription of ribosomal genes and de-novo protein synthesis. As this was observed in both human and zebrafish cells, it is likely that the lack of correlation has been evolutionarily conserved.”

and

“Furthermore, our analysis of data obtained from human HSCs and progenitors revealed that ribosomal gene expression levels are highly similar between the different progenitor types and stem cells, despite their significantly different protein synthesis rates (Signer et al., 2014).”

Figure 7: Heatmap of ribosomal gene expression in human HSCs and progenitors. Clustering of all human ribosomal genes across different HSPC populations, using Euclidean distance and Ward Linkage.

Figure 8: Correlation analysis of expressed ribosomal genes across 891 HSPCs. Pairwise Pearson correlation revealed similar expression levels of the expressed cytosolic ribosomal genes in HSC and various progenitors (MPP, CMP, GMP, MEP and MLP).

4, A minor point, how was the normalization of gene expression against total level of transcripts performed? It is important to know whether any changes in the total transcript levels at the single cell level and how they affect the identification of variable genes used to define the cell states. The reviewer can't seem to find such information in the text or methods.

> We apologise for omitting this information from the manuscript. In the first step, we obtained the total number of counts for each gene in each cell using Sailfish. We next divided the number of counts for each gene by the total number of counts (i.e. sum of all counts per cell) in each cell. Then, we multiplied the acquired number by 1,000,000 to generate counts per million (CPM). CPMs were further processed using size-factor normalisation method (by Lun et al. 2016) which is specifically designed for scRNA-seq data, to eliminate all other sources of noise in single-cell data.

Reviewer #2 (Remarks to the Author):

In this study, Athanasiadis et al use single cell RNA-sequencing to transcriptionally define zebrafish hematopoiesis. Computational construction of the blood lineage tree suggests that lineage commitment of hematopoietic stem and progenitor cells occurs through a single path and intermediate progenitor. The authors further demonstrate that lineage commitment is associated with transcriptional repression of ribosomal genes. In some aspects, this study is consistent with several other transcriptional profiling and functional studies that have defined the presence of unipotent myeloid and erythroid progenitors. Unfortunately, this study seems significantly underpowered to faithfully reconstruct the complexity of native hematopoiesis.

1. The central branch of the reconstructed hematopoietic tree contains hematopoietic stem and progenitor cells. Despite the transcriptional clustering, this is a functionally heterogeneous population of cells, and likely contains very few true HSCs. Transplantable HSCs are present in the zebrafish kidney at a frequency of $\sim 1/38,000$ (Hess et al., PNAS 2013; 110:4327). To what extent are HSCs enriched by the use of transgenic zebrafish in this study? How many of 1,422 sequenced cells are likely to be true HSCs?

> As pointed out by the Reviewer, the middle part of our tree is indeed heterogeneous and contains a few HSCs. Based on previous studies it has been shown that around 1:35 runx1+ cells are true, transplantable HSC (Tamplin et al., 2015). In the initial data set we sequenced 160 runx1+ cells, therefore the expected number of true HSCs using just this line would be around 4-5 cells. However, our analysis also included cd41^{dim} cells, which have been shown to contain transplantable HSCs as well (Ma et al., 2011). So the total number of HSCs was most probably higher than 5 cells.

In the revised manuscript, we have included an additional 302 runx1+ cells in our analysis, making a total of 462 analysed runx1+ cells. (Of these, 153 runx1+ cells were newly sorted and sequenced and another 149 runx1+ cells were previously sequenced but not included in the initial analysis). Therefore, the expected number of HSCs just from this line is around 13 cells. To predict the potential HSCs in the whole middle part of our tree we computed the S^{rel} index, calculated using the Kullback–Leibler distance of the predicted probabilities compared to the expected one, for each of the four different branches (Velten et al., 2017). Based on this prediction we identified 35 HSCs in our data set. Of all cells that were predicted to be HSC, 14 were runx1 positive which is in the same order of magnitude as we would expect based on frequency of HSC in runx1 line (as calculated based on transplantation experiments in Tamplin et al., 2015). Importantly, the majority of predicted HSCs were runx1+ and cd41^{dim} cells, both of which have been previously identified to contain transplantable HSC (Tamplin et al., 2015 and Ma et al., 2011) lending further confidence to our computational prediction.

Figure 6. Transcriptionally similar cells display different probabilities of being stem cells. a) Cells predicted to be stem cells in the middle part of the lineage tree according to their stemness index. The insert shows the new Monocle tree including transplanted cells (1,724 single cells and 1,871 highly variable genes). b) Distribution of stemness scores in different branches of the tree showing the presence of potential HSCs exclusively in the HSPC branch. c) Contribution of different transgenic lines to predicted stem cells.

2. The likely absence of significant numbers of HSCs complicates the interpretation of the ribosomal gene expression data. Studies demonstrating low rates of protein synthesis within HSCs were performed using highly purified cell populations (based on functionally validated cell surface marker profiles). These same studies demonstrated that many progenitors have very high rates of protein synthesis. Since the HSPC branch in the present study does not distinguish between HSCs and progenitors, and HSCs are likely to be an exceedingly rare contributor to this population, the relatively high expression of ribosomal genes is most simply explained as a consequence of the high progenitor content of this population.

> We agree with the Reviewer that the previous study (Signer et al., 2014) that demonstrated low rates of protein synthesis within HSCs used highly purified cell populations as defined by cell surface markers. Considering that the middle part of our reconstructed tree contains a small fraction of HSCs and significant number of progenitors, one of the plausible explanations (as the Reviewer pointed out) for the high expression of ribosomal genes in the middle part of our tree is the high progenitor content.

>To address this issue, we decided to check the level of ribosomal expression in human single-cell RNA-Seq data in phenotypically well-defined cell populations (HSC, MPP, CMP, GMP, MEP and MLP). We used a human single-cell RNA-Seq data set (deposited in the Gene Expression Omnibus (GEO) under accession code GSE75478 and published (Velten et al., 2017). The analysis revealed very strong correlations (0.92-0.99) in the expression of the ribosomal genes between the HSCs and all five progenitor populations (Fig. 8). Therefore, even though HSCs have

higher de novo protein synthesis (Signer et al., 2014) than any of the progenitor populations they do not have lower transcription of genes that encode ribosomal proteins. For further details please see “Summary of the additional data and analyses ” , page 5, point 4.

We would like to point out that we do not challenge previous findings related to the higher protein synthesis in progenitor cells compared to HSC. We just demonstrated that even in the phenotypically well-defined cell populations the level of transcription of ribosomal genes in HSCs and progenitor populations is strongly correlated, which is in line with the observation we had in the middle part of our reconstructed tree in zebrafish.

Modified text (pages 7 and 10):

“A previous study showed that the rate of protein synthesis in murine HSCs is considerably lower than that of progenitor populations (i.e. CMPs, GMPs and MEPs) (Signer et al 2014). This is not in line with our transcriptional analysis, which showed a decrease in ribosomal gene expression during differentiation. In order to address this discrepancy, we considered the correlation in ribosomal gene expression between human phenotypic HSCs (CD34+CD38-CD45RA-CD90+CD49f+) and the different progenitor fractions (for details please see Methods). We used a publicly available scRNA-Seq data set from bone marrow derived HSPCs and analysed the expression of genes that encode cytosolic ribosomal proteins. After calculating average log10 expression profiles for each of the six different cell types (HSC, MPP, MLP, CMP, GMP and MEP), we calculated the pairwise Pearson correlation. The analysis revealed very strong correlations (0.92-0.99) between the the ribosomal gene expression in HSCs and all five progenitor populations (Supplementary Fig. 9). Therefore, even though HSCs have 10-fold higher de novo protein synthesis (Signer et al., 2014) their level of expression of genes that encode ribosomal proteins is similar (highly correlated) to that of the progenitor populations. Our results described above suggested that there is a poor correlation between the level of transcription of ribosomal genes and de-novo protein synthesis. As this was observed in both human and zebrafish cells, it is likely that the lack of correlation has been evolutionarily conserved.”

and

“Furthermore, our analysis of data obtained from human HSCs and progenitors revealed that ribosomal gene expression levels are highly similar between the different progenitor types and stem cells, despite their significantly different protein synthesis rates (Signer et al 2014).”

3. The observation that ribosome gene expression decreases as cells differentiate is not unexpected. Ribosome biogenesis is a key feature of cell proliferation, and rapidly dividing progenitors would depend upon ribosome production more than terminally differentiated cells with limited (or no) proliferative potential. How does ribosome gene expression correlate with cell cycle gene expression?

> This is a very good point, so we checked if there is a correlation between ribosomal gene expression and cell cycle gene expression. In the first step, we identified a total of 342 zebrafish genes which are annotated as “GO:0007049” i.e. “cell cycle” using BioMart (version 83). Next, we performed clustering using only those cell cycle related genes (which are expressed in more than 10% of cells in each of the branches) and dynamically expressed ribosomal genes. Overall, we did not see a correlation/anticorrelation between ribosomal gene expression and cell cycle gene expression across the differentiation process (Fig. 10). In all cases, dynamically expressed ribosomal genes were clustered together with high correlation values (Fig. 10a), suggesting that ribosomal genes were co-expressed in a similar way across the pseudotime (Fig. 10b and c). In contrast, genes related to the cell cycle clustered together with very low correlation values (Fig. 10a), suggesting random gene expression (Fig. 10b and c). The latter is more clear in Figure 10b, where expressions across pseudotime for each of the four branches are illustrated. Ribosomal gene expression decreased across the pseudotime, whereas cell cycle genes switched on/off in pseudotime. Average expression patterns of ribosomal and cell cycle genes again identified two main trends - one showing decreased expression of ribosomal genes and the second showing low, nearly constant expression of cell cycle genes (Fig 10c).

Figure 10: Correlation analysis between ribosomal and cell cycle related genes. (a) Correlation heatmaps across all ribosomal and cell cycle genes, (b) correlation heatmaps of

ribosomal and cell cycle genes in pseudotime and (c) average expression patterns of ribosomal and cell cycle genes in pseudotime.

Modified text (pages 7 and 10):

“Importantly, the observed downregulation of ribosomal genes in pseudotime was not correlated with the cell cycle state of the cell, apart from a weak correlation in the erythrocytic lineage (Supplementary Fig. 8).”

and

“Importantly, this change was not linked to the expression of cell cycle specific genes, excluding proliferation rates as a potential reason for these data.”

4. Why are there no lymphoid lineage cells present in this reconstruction? Were they deliberately excluded?

> We did not deliberately exclude the lymphoid lineage. At the time when we started this study, we did not have a zebrafish line that would enrich for B- and NK cells. We did use the cd4 transgenic line in our study, but these cells mature in the thymus rather than in the kidney and more mature cells were therefore not captured. As a result, cd4 cells present in the kidney clustered, as expected, in the HSPC part of the reconstructed tree and in monocyte branch.

Modified text (page 4):

“Mature lymphocytes could not be detected, most likely as T-cells mature in the thymus and B-cells are comparatively rare and were not enriched for.”

Reviewer #3 (Remarks to the Author):

This manuscript presents the transcriptional profiles of 1,422 zebrafish haematopoietic cells by single-cell RNA sequencing (scRNA-seq) technology. The authors performed quite a few scRNA-seq analyses, including quality control, gene quantification, cell subtyping, differentiation trajectory mapping, differential expression analysis between different defined cell types. Then the authors reconstructed the blood lineage tree, defined 5 distinct cell “states”, investigated the differential expression patterns between each state and identified a pool of state-specific expression marker genes. In addition, they also compared the expressions of ribosomal genes and state-specific genes during lineage differentiation, analyzed the conservation of those state-specific genes between zebrafish and mammals, and built a cloud repository, BASiCz, for data visualization. Overall, this study provides a fantastic resource for dissecting haematopoiesis at the transcriptome level.

In this study, the authors performed 1,422 scRNA-seq analyses for zebrafish haematopoietic cells, comprehensively investigated the differentiation trajectories and state-specific expression patterns, built a cloud repository for the data visualization, and finally analyzed the conservation of haematopoietic program between zebrafish and mammals. Considering that this manuscript is well-organized and clearly-written, I only give two minor comments:

1. In the scRNA-seq data processing, I noted that the authors used Sailfish (without alignment) tools to quantify gene expression. However, uniquely-aligned reads are commonly recommended

for scRNA-seq data processing (Grun, D. and A. van Oudenaarden (2015). Cell 163(4): 799-810.). Have authors compared two different processing strategies (“alignment + quantification with uniquely-mapped reads” vs. “quantification without alignment”)? How does it affect the accuracy of gene quantification?

> In order to quantify our scRNA-seq dataset with uniquely-mapped reads, we re-aligned our reads using STAR (Dobin et al., 2013) software (Version STAR-2.5.3a). We set the parameters `--outSAMmultNmax 1` to output exactly one SAM line for each mapped read and `--outSAMtype BAM Unsorted` to further quantify the BAM files using Salmon (Patro et al., 2017)(Version Salmon-0.8.2) The setting `-l IU` for pair-end library was used. Finally, the same reference genome was used (as described for the Sailfish approach) so that results are comparable.

Gene quantification accuracy was assessed by calculating the average \log_{10} TPM+1 gene expressions for each cell of the 21 different sequenced plates reported by Salmon and Sailfish, after removing ERCCs and rescaling TPMs to million. We only used wells with single cells. Data points from quality control wells (i.e. wells without cells and with multiple cells) were removed. Pearson correlation of the average gene expressions between Salmon and Sailfish ranged from 0.81 to 0.91 suggesting strong correlation between alignment that included uniquely-mapped reads and those that did not (Figure 2). As expected, the number of detected genes (TPM > 1) was lower for Salmon compared to Sailfish (Figure 3). However, the gene’s variability distribution (Coefficient of Variation CV) across single cells for each plate was comparable between the two methods (Figure 4).

Figure 2. Pearson correlation of the average gene expressions between Salmon and Sailfish. The gene quantification accuracy was assessed by selecting the average \log_{10} TPM+1 gene expression value for each of the 21 sequenced plates.

Figure 3. Violin plots of the number of detected genes (TPM>1) at single-cell level. Salmon (unique mapped reads) and Sailfish (multi-mapped reads) were compared for each of the 21 plates.

Figure 4. Distribution of the Coefficient of Variation. Comparison of $\log_{10}(CV+0.001)$ gene expression (TPMs) values at a single-cell level between Salmon (unique mapped reads) and Sailfish (multi-mapped reads), across 21 plates.

2. Please cite the papers of software/tools used in this study if applicable, such as the pcaMethods R package used in this study.

> We have included the requested information in the Methods section of the manuscript for the pheatmap, cluster, FastICA, pcaMethods, ggplot2, Tophat2, Bowtie2, Kallisto, htseq, cufflinks, shiny and plotly software/tools.

>We have also updated the References section accordingly.

“Kolde, R. pheatmap: Pretty Heatmaps. R package version 1.0.8 edn (2015).”

“Maechler, M., Rousseeuw, P., Struyf, A., Hubert, M. & Hornik, K. cluster: Cluster Analysis Basics and Extensions. R package version 2.0.5 edn (2017).”

“Marchini, J.L., Heaton, C. & Ripley, B.D. fastICA: FastICA Algorithms to perform ICA and Projection Pursuit. R package version 1.2-0 edn (2013).”

“Stacklies, W., Redestig, H., Scholz, M., Walther, D. & Selbig, J. pcaMethods - A bioconductor package providing PCA methods for incomplete data. *Bioinformatics* **23**, 1164-1167 (2007).”

“Wickham, H. ggplot2: Elegant Graphics for Data Analysis. R package version 2.2.1 edn (Springer-Verlag New York, 2009).”

“Kim, D. *et al.* TopHat2: accurate alignment of transcriptomes in the presence of insertions, deletions and gene fusions. *Genome Biol.* **14**, R36 (2013).”

“Langmead, B. & Salzberg, S. L. Fast gapped-read alignment with Bowtie 2. *Nat. Methods* **9**, 357–359 (2012).”

“Bray, N. L., Pimentel, H., Melsted, P. & Pachter, L. Near-optimal probabilistic RNA-seq quantification. *Nat. Biotechnol.* **34**, 525–527 (2016).”

“Anders, S., Pyl, P. T. & Huber, W. HTSeq--a Python framework to work with high-throughput sequencing data. *Bioinformatics* **31**, 166–169 (2015).”

“Trapnell, C. *et al.* Transcript assembly and quantification by RNA-Seq reveals unannotated transcripts and isoform switching during cell differentiation. *Nat. Biotechnol.* **28**, 511–515 (2010).”

“Chang, W., Cheng, J., Allaire, J. J., Xie, Y. & McPherson, J. shiny: Web Application Framework for R. (2017).”

“Sievert, C. *et al.* plotly: Create Interactive Web Graphics via ‘plotly.js’. (2017).

References

Dobin, A. *et al.* STAR: Ultrafast universal RNA-seq aligner. *Bioinformatics* **29**, 15-21 (2013).

Ma, D., Zhang, J., Lin, H.F., Italiano, J. & Handin, R.I. The identification and characterization of zebrafish hematopoietic stem cells. *Blood* **118**, 289-297 (2011).

Patro, R., Duggal, G., Love, M.I., Irizarry, R.A. & Kingsford, C. Salmon provides fast and bias-aware quantification of transcript expression. *Nature Methods* (2017).

Robert, C. & Watson, M. Errors in RNA-Seq quantification affect genes of relevance to human disease. *Genome Biology* **16** (2015).

Signer, R.A., Magee, J.A., Salic, A. & Morrison, S.J. Haematopoietic stem cells require a highly regulated protein synthesis rate. *Nature* **509**, 49-54 (2014).

Tamplin, O.J. *et al.* Hematopoietic Stem Cell Arrival Triggers Dynamic Remodeling of the Perivascular Niche. *Cell* **160**, 241-252 (2015).

Velten, L. *et al.* Human haematopoietic stem cell lineage commitment is a continuous process. *Nat Cell Biol* (2017).

Reviewers' comments:

Reviewer #1 (Remarks to the Author):

The revised version of the manuscript is greatly improved with respect to the original version. Specifically, authors have examined the robustness of the reconstruction of the lineage tree using variable genes defined by different cutoffs, and different alignment strategies of the scRNA-seq data. Authors have also included additional runx1+ single cells to increase the number of possibly true HSCs in the originally defined HSPC population. The new results have improved the clarity and rigor of the original analyses. However, the main concern remains for the interpretation of the ribosomal gene expression data related to protein synthesis in HSCs vs differentiated lineages. Specifically, even with the addition of 302 runx1+ cells to the analysis, the possibly true HSCs are still rare (e.g. predicted 35 of 214 cells in the reconstructed HSPC population, Fig. 4) and constitute only a very minor portion of this functionally heterogeneous population of cells. Given the nature of the heterogeneity and relatively low resolution of the "HSPC state" in this study, it remains questionable whether one can compare the findings with previous studies performed with functionally validated and highly purified populations of cells. In the referenced Signer et al. study, it was noted that de novo protein synthesis rate was extremely low in HSCs and MPPs (based on functionally validated cell surface markers), markedly increased in progenitor populations (CMP, GMP and MEP), and then decreased to various degree in differentiated lineages. Therefore, the observed downregulation of ribosomal gene expression associated with lineage differentiation in this study may be related to the downregulation of protein synthesis between progenitor populations and differentiated lineages as previously described. Moreover, it should be noted that authors inferred the relevance to protein synthesis based on the mRNA expression levels of ribosomal genes instead of the direct measurement of de novo protein synthesis rate. Given these confounding factors, the authors are advised to significantly revise the relevant interpretation and discussion of these results.

A minor point, it was repeatedly stated in the main text and the rebuttal letter that "...even though HSCs have 10 fold higher de novo protein synthesis..." (e.g. pages 6, 11 and 12 of the rebuttal letter), should it be "lower"?

Reviewer #2 (Remarks to the Author):

Athanasiadis et al have done a good job addressing some of the technical issues in this study, but some of my initial concerns still remain.

1. The number of HSCs sequenced is still only estimated to be 35, which seems rather low. At what number of cells is this analysis properly powered?

2. To address issues with the ribosomal gene expression data, the authors included the following paragraph in the revised manuscript:

"A previous study showed that the rate of protein synthesis in murine HSCs is considerably lower than that of progenitor populations (i.e. CMPs, GMPs and MEPs)³⁵. This is not in line with our transcriptional analysis, which showed a decrease in ribosomal gene expression during differentiation. In order to address this discrepancy, we considered the correlation in ribosomal gene expression between human phenotypic HSCs (CD34+ CD38- CD45RACD90+ CD49f+) and the different progenitor fractions (for details please see Methods). We used a publicly available scRNA-Seq data set from bone marrow derived HSPCs and analysed the expression of genes that encode cytosolic ribosomal proteins. After calculating average log₁₀ expression profiles for each of the six different cell types (HSC, MPP, MLP, CMP, GMP and MEP), we calculated the pairwise Pearson correlation. The analysis revealed very strong correlations (0.92-0.99) between the ribosomal gene expression in HSCs and all five progenitor populations (Supplementary Fig. 9).

Therefore, even though HSCs have 10-fold higher de novo protein synthesis³⁵, their level of expression of genes that encode ribosomal proteins is similar (highly correlated) to that of the progenitor populations. Our results described above suggested that there is a poor correlation between the level of transcription of ribosomal genes and de novo protein synthesis. As this was observed in both human and zebrafish cells, it is likely that the lack of correlation has been evolutionarily conserved.”

a. Are progenitor populations such as CMP, GMP, and MEP contained within the central HSPC branch of the computationally constructed lineage tree? If they are, then the conclusion should be that hematopoietic progenitors downregulate ribosomal gene expression as they terminally differentiate. Because the central branch is heterogeneous and contains mostly progenitors, it should not be considered representative of stem cells.

b. The results described are for transcriptional profiles in humans and zebrafish, and the protein synthesis data referenced are from mouse. To conclude that there is poor correlation between transcription of ribosomal genes and protein synthesis these analyses should probably be done in the same species.

c. Does the correlation between ribosome gene expression between cell populations conclusively demonstrate that the absolute transcript number is similar? How has expression been normalized?

d. The paragraph states that “HSCs have 10-fold higher de novo protein synthesis...” It should say lower de novo protein synthesis.

e. The Signer et al Nature 2014 study reported that ribosomal RNA expression did not significantly differ between HSCs and progenitors. This would appear to be consistent with the conclusion that cell-type specific differences in protein synthesis are not primarily caused by differences ribosomal gene expression.

3. Is the use of a cell cycle gene set to correlate proliferation and ribosome gene expression appropriate? Does the cell cycle gene set only contain positive regulators or markers of actively cycling cells or does it also contain negative regulators? Ribosome gene expression should be measured within cycling and non-cycling cells to properly address this issue.

Reviewer #3 (Remarks to the Author):

In this revised manuscript, the authors included more analyses and solved the previous concerns for their scRNA-seq analysis pipeline. The authors also included 300 new single cells from transplantation recipients, increasing the significance of this work as a resource for the community. Overall, this work provides a very good example for scRNA-seq data analysis and enhances the understanding of haematopoietic program by single-cell transcriptomics. I recommend "accept"

We would like to thank all three Reviewers for their valuable feedback. We were able to address most issues with changes to the wording of the manuscript. Furthermore, we carried out some additional computational analysis to show the similarity in ribosomal gene expression between assigned HSCs and the remaining progenitor population. A summary of our new analysis is outlined on top of the document, with detailed responses to the Reviewers below. Our responses to the comments are in *italics* and direct quotes from the updated manuscript are in blue.

Summary of the additional analyses

1) Comparing ribosomal gene expression between potential zebrafish HSCs and more mature progenitors

To ensure the similarity of ribosomal gene expression between cells classified as HSCs and the other progenitors, we calculated the Pearson correlation between HSCs and the other progenitors (Figure 1). This analysis revealed a high correlation between the populations (0.986).

Figure 1: Pearson correlation between cells classified as HSCs and the remaining progenitor population.

Next, we investigated whether any individual genes were behaving differently from the whole population. To do that, we generated a heatmap showing the gene expression of individual ribosomal genes across the HSC and progenitor populations (Figure 2). This analysis did not reveal any obvious trends. To ensure that this result was not due to all genes fluctuating in the same way (i.e. a general decrease or increase), we also calculated the coefficient of variation (CV) of ribosomal transcripts (Figure 2) using the absolute

expression values. Results suggest that there was low level of fluctuations of the absolute expression values across the HSC and progenitors cells, since the mean CV was 0.180 ± 0.033 .

Figure 2: Ribosomal gene expression across predicted HSCs and the remaining progenitors. Coefficient of variation across single cells for each ribosomal gene is also presented.

Please note that the figures presented for this analysis have not been included in the revised version of the manuscript, but the statistical results have been.

1) Comparing ribosomal gene expression between the different human HSC and progenitor populations

To address the concern of Reviewer#2 that the absolute expression of ribosomal genes might differ between different human progenitor populations, we calculated the Coefficient of Variation (CV) for each ribosomal gene across the six different cell types. The CV is a measure of relative variability and can be used to quantify to what extent expression values of ribosomal genes differ in different cell types. Our results suggest that for the case of cytosolic ribosomal genes, absolute gene expression values across different cell types showed low levels of fluctuation (CV score 0.116 ± 0.047), whereas mitochondrial ribosomal genes were randomly expressed at different levels (CV score 1.589 ± 1.033). We have added this data to the heatmap previously presented in the manuscript (Figure 3a). Together with the high Pearson correlations between the different fractions (Figure 3b), these data conclusively show that ribosomal gene expression is at similar levels in the different progenitor populations.

Figure 3. Analysis of ribosomal genes in human HSCs and progenitors. a) Heatmap of ribosomal gene expression in human HSCs and progenitors. Clustering of all human ribosomal genes across different HSPC populations, using Euclidean distance and Ward Linkage. On the top, cytosolic ribosomal genes are shown, whereas the bottom cluster consists of mitochondrial ribosomal genes. On the left hand side, the calculated coefficient of variation (CV) across cells is presented for each gene. b) Correlation analysis of expressed ribosomal genes across 891 HSPCs. Pairwise Pearson correlation revealed similar expression levels of the expressed cytosolic ribosomal genes in HSC and various progenitors (MPP, CMP, GMP, MEP and MLP).

This figure has been included in the manuscript as Supplementary Figure 9.

REVIEWERS' COMMENTS:

Reviewer #1 (Remarks to the Author):

The revised version of the manuscript is greatly improved with respect to the original version. Specifically, authors have examined the robustness of the reconstruction of the lineage tree using variable genes defined by different cutoffs, and different alignment

strategies of the scRNA-seq data. Authors have also included additional runx1+ single cells to increase the number of possibly true HSCs in the originally defined HSPC population. The new results have improved the clarity and rigor of the original analyses. However, the main concern remains for the interpretation of the ribosomal gene expression data related to protein synthesis in HSCs vs differentiated lineages. Specifically, even with the addition of 302 runx1+ cells to the analysis, the possibly true HSCs are still rare (e.g. predicted 35 of 214 cells in the reconstructed HSPC population, Fig. 4) and constitute only a very minor portion of this functionally heterogeneous population of cells. Given the nature of the heterogeneity and relatively low resolution of the “HSPC state” in this study, it remains questionable whether one can compare the findings with previous studies performed with functionally validated and highly purified populations of cells. In the referenced Signer et al. study, it was noted that de novo protein synthesis rate was extremely low in HSCs and MPPs (based on functionally validated cell surface markers), markedly increased in progenitor populations (CMP, GMP and MEP), and then decreased to various degree in differentiated lineages. Therefore, the observed downregulation of ribosomal gene expression associated with lineage differentiation in this study may be related to the downregulation of protein synthesis between progenitor populations and differentiated lineages as previously described.

Signer et al indeed observed lower rates of protein synthesis in GR-1+ myeloid cells than in CMPs, GMPs and MEPs, suggesting that protein synthesis is decreased in fully mature cells compared to progenitors. However, in the aforementioned study HSCs and MPPs still had lower protein synthesis rates than GR-1+ cells and lymphoid cells.

Our comparison of published human HSCs and progenitors suggests no difference in ribosomal gene expression between HSCs and progenitors. This would be equivalent to the middle branch of our tree. In addition, our zebrafish data show a decrease in the expression of ribosomal genes in committed cells (cells in four branches) as they differentiate. These observations show that the expression of ribosomal genes in HSPCs does not strictly follow the levels of protein synthesis, as reported in the mouse model.

Furthermore, we have carried out additional analysis to ensure that ribosomal gene expression between cells classified as HSCs and other progenitors is similar. First, we calculated the Pearson correlation between HSCs and the other progenitors (Figure 1). This analysis revealed a high correlation between the populations (0.986).

Figure 1: Pearson correlation between cells classified as HSCs and the remaining progenitor population.

Next, we investigated whether any individual genes were behaving differently from the whole population. To do that, we generated a heatmap showing the gene expression of individual ribosomal genes across the HSC and progenitor populations (Figure 2). This analysis did not reveal any obvious trends.

Figure 2: Ribosomal gene expression across predicted HSCs and the remaining progenitors. Coefficient of variation across single cells for each ribosomal gene is also presented.

Modified text, page 7:

“Next, we compared ribosomal gene expression between the predicted HSCs and the remaining progenitors in the middle branch. The absolute number of ribosomal transcript was similar between the two populations (CV score is 0.180 ± 0.033). In agreement with this, there was a high Pearson correlation (0.986) between predicted HSCs and progenitors. Overall, this suggests highly similar ribosomal gene expression between HSCs and more committed progenitors.”

Since transcriptional analysis was done using human and zebrafish cells, whereas protein synthesis was studied in the mouse, we have removed references to the murine data where appropriate.

Moreover, it should be noted that authors inferred the relevance to protein synthesis based on the mRNA expression levels of ribosomal genes instead of the direct measurement of de novo protein synthesis rate. Given these confounding factors, the authors are advised to significantly revise the relevant interpretation and discussion of these results.

We agree that our new data is limited to gene expression rather than direct measurements of protein synthesis. We have rewritten the section as outlined above to account for this.

A minor point, it was repeatedly stated in the main text and the rebuttal letter that "...even though HSCs have 10 fold higher de novo protein synthesis..." (e.g. pages 6, 11 and 12 of the rebuttal letter), should it be "lower"?

We apologise for the mistake and have removed the passage from the text.

Reviewer #2 (Remarks to the Author):

Athanasiadis et al have done a good job addressing some of the technical issues in this study, but some of my initial concerns still remain.

1. The number of HSCs sequenced is still only estimated to be 35, which seems rather low. At what number of cells is this analysis properly powered?

RNA-seq is routinely used to successfully identify extremely rare cell populations (Proserpio and Lönnberg 2015), where clusters consisting of fewer than ten cells are common. Therefore, we consider the number of HSCs in our study to be sufficient.

2. To address issues with the ribosomal gene expression data, the authors included the following paragraph in the revised manuscript:

"A previous study showed that the rate of protein synthesis in murine HSCs is considerably lower than that of progenitor populations (i.e. CMPs, GMPs and MEPs)³⁵. This is not in line with our transcriptional analysis, which showed a decrease in ribosomal gene expression during differentiation. In order to address this discrepancy, we considered the correlation in ribosomal gene expression between human phenotypic HSCs (CD34+ CD38- CD45RACD90+ CD49f+) and the different progenitor fractions (for details please see Methods). We used a publicly available scRNA-Seq data set from bone marrow derived HSPCs and analysed the expression of genes that encode cytosolic ribosomal proteins. After calculating average log₁₀ expression profiles for each of the six different cell types (HSC, MPP, MLP, CMP, GMP and MEP), we calculated the pairwise Pearson correlation. The analysis revealed very strong correlations (0.92-0.99) between the the ribosomal gene expression in HSCs and all five progenitor populations (Supplementary Fig. 9). Therefore, even though HSCs have 10-fold higher de novo protein synthesis³⁵, their level of expression of genes that encode ribosomal proteins is similar (highly correlated) to that of the progenitor populations. Our results described above suggested that there is a poor correlation between the level of transcription of ribosomal genes and de novo protein synthesis. As this was observed in both human and zebrafish cells, it is likely that the lack of correlation has been evolutionarily conserved."

a. Are progenitor populations such as CMP, GMP, and MEP contained within the central HSPC branch of the computationally constructed lineage tree? If they are, then the conclusion should be that hematopoietic progenitors downregulate ribosomal gene expression as they terminally differentiate. Because the central branch is heterogeneous and contains mostly progenitors, it should not be considered representative of stem cells.

Due to the way the haematopoietic tree was reconstructed in our study (i.e. without any cell-surface markers), we cannot make any conclusive claims on how closely our progenitors would resemble phenotypically/marker defined populations in mouse or human. Our DNN analysis revealed the presence of 35 potential HSCs in the middle branch, suggesting that the remaining 179 cells in this branch are progenitors at various stages of maturation, so it is true that the branch contains mostly progenitors. However, our analysis showed that these progenitors are transcriptionally very similar to true HSCs, hence why they cluster together.

b. The results described are for transcriptional profiles in humans and zebrafish, and the protein synthesis data referenced are from mouse. To conclude that there is poor correlation between transcription of ribosomal genes and protein synthesis these analyses should probably be done in the same species.

Ideally, we would have used the same species for this analysis. However, currently the identity of the HSCs and different progenitors in zebrafish can only be resolved following single cell transcriptional analysis. In other words, since there are no antibodies to specifically sort distinct populations of HSPCs, it is not possible to compare their protein synthesis.

We analysed previously published human data just to show the similarity between findings in zebrafish and higher vertebrates, such as humans. One can indeed argue that the process is not conserved in mouse and that the expression of ribosomal genes follows a different trend in this species.

We have therefore reworded these parts of the manuscript as described above.

c. Does the correlation between ribosome gene expression between cell populations conclusively demonstrate that the absolute transcript number is similar? How has expression been normalized?

In addition to the correlations shown in Figure S9b, we used the heatmap shown in Figure S9a to visualise the absolute expression of the ribosomal genes. As can be seen, absolute expression values are very similar across all cell types. Taken together, these data unequivocally show that absolute transcript numbers are comparable across the different fractions. Additionally, we have added the Coefficient of Variation (CV) for each ribosomal gene across the six different cell types. Our results suggest that for the case of cytosolic ribosomal genes, absolute gene expression values across different cell types showed low levels of fluctuation (CV score 0.116 ± 0.047), whereas mitochondrial ribosomal genes were randomly expressed at different levels (CV score 1.589 ± 1.033). We have added this data to the heatmap previously presented in the manuscript (Figure 4a). Together with the high Pearson correlations between the different fractions (Figure 4b), these data conclusively show that ribosomal gene expression is at similar levels in the different progenitor populations.

Figure 3. Analysis of ribosomal genes in human HSCs and progenitors. a) Heatmap of ribosomal gene expression in human HSCs and progenitors. Clustering of all human ribosomal genes across different HSPC populations, using Euclidean distance and Ward Linkage. On the top, cytosolic ribosomal genes are shown, whereas the bottom cluster consists of mitochondrial ribosomal genes. On the left hand side, the calculated coefficient of variation (CV) across cells is presented for each gene. b) Correlation analysis of expressed ribosomal genes across 891 HSPCs. Pairwise Pearson correlation revealed similar expression levels of the expressed cytosolic ribosomal genes in HSC and various progenitors (MPP, CMP, GMP, MEP and MLP).

Modified text, page 8:

“To quantitatively assess whether the absolute expression value of each gene fluctuates, we calculated the Coefficient of Variation (CV)³⁶ for each ribosomal gene across the six different cell types. Our results suggest that for the case of cytosolic ribosomal genes, absolute gene expression values across different cell types showed low levels of fluctuation (CV score 0.116 ± 0.047), whereas mitochondrial ribosomal genes were randomly expressed at different levels (CV score 1.589 ± 1.033). This shows that ribosomal gene expression of human HSCs is highly similar to more mature progenitors, confirming an evolutionary conservation of this trend from zebrafish.”

No additional normalisation was performed, other than what was done to all data to receive normalised counts per million. We have slightly clarified the methodology in the text.

Modified text, page 15:

“After normalisation with the scran package of the resulting CPMs (similarly to zebrafish data), [...]”

d. The paragraph states that “HSCs have 10-fold higher de novo protein synthesis...” It should say lower de novo protein synthesis.

We apologise for the mistake and have removed the passage from the text.

e. The Signer et al Nature 2014 study reported that ribosomal RNA expression did not significantly differ between HSCs and progenitors. This would appear to be consistent with the conclusion that cell-type specific differences in protein synthesis are not primarily caused by differences ribosomal gene expression.

We agree with the reviewer that the differences in protein synthesis are not primarily driven by differences in ribosomal RNA expression. We expanded the findings reported by Signer et al by including the non-RNA components of the ribosome in our analysis.

3. Is the use of a cell cycle gene set to correlate proliferation and ribosome gene expression appropriate? Does the cell cycle gene set only contain positive regulators or markers of actively cycling cells or does it also contain negative regulators? Ribosome gene expression should be measured within cycling and non-cycling cells to properly address this issue.

We agree that the choice of which ribosomal genes to focus on is vital for a correct interpretation of the experiments. Because of that, we used the broadest annotation available (GO:0007049 “cell cycle”), which contains both positive and negative regulators (please see <http://www.ebi.ac.uk/QuickGO/GTerm?id=GO:0007049> for details). We believe this is the most appropriate way to analyse our data, as it avoids bias introduced by selecting specific subsets.

We would like to point out that the order of the cells on the reconstructed trajectory was determined by their differentiation state rather than their cell cycle state, meaning that dividing and non-dividing cells at similar developmental stages were intermixed.

Within our reconstructed trajectory, we reported down-regulation of ribosomal genes in four branches. Focusing further on measuring ribosomal gene expression between cycling and non-cycling cells at the same stage of differentiation is beyond the scope of this manuscript. Also, there are many technical challenges; for example the tree was reconstructed based on the transcriptome of cells (i.e. after the cells were lysed and sequenced). We do not have a way of isolating specific subsets of cells which are at the same differentiation state but different cycling state.

Reviewer #3 (Remarks to the Author):

In this revised manuscript, the authors included more analyses and solved the previous concerns for their scRNA-seq analysis pipeline. The authors also included 300 new single cells from transplantation recipients, increasing the significance of this work as a resource for the community. Overall, this work provides a very good example for scRNA-seq data analysis and enhances the understanding of haematopoietic program by single-cell transcriptomics.

I recommend "accept"

References

Proserpio, V. and Lönnberg, T., 2016. Single-cell technologies are revolutionizing the approach to rare cells. Immunology and cell biology, 94(3), p.225.

REVIEWERS' COMMENTS:

Reviewer #1 (Remarks to the Author):

The authors have further revised the manuscript related to the results on ribosomal gene expression in HSPCs and differentiated lineages. They also included additional analysis on the correlation of ribosomal gene expression between HSCs and progenitor cells based on scRNA-seq in zebrafish and human. These revisions have greatly improved the clarity of the main findings, and adequately addressed the reviewer's original comments. The rich resource of scRNA-seq data and the new insights into the regulation of hematopoiesis in zebrafish with complementary analysis from mouse and human provided by this study should stimulate interests in follow-up data analysis and functional studies.

Reviewer #2 (Remarks to the Author):

Overall this manuscript is much improved. I recommend that the manuscript be accepted, but that the authors should tone down language that risks over-interpreting the data regarding ribosomal gene expression within HSCs. It is fair to conclude that ribosomal gene expression is reduced during terminal differentiation, and that ribosomal gene expression in HSCs is more similar to progenitors than differentiated cells. In general, these observations are novel and support the conclusion that cell-type specific differences in protein synthesis are not regulated by differences in ribosomal gene expression. However, in the absence of being able to isolate and analyze purified HSCs and in the absence of spike-in controls that correct for absolute mRNA content between cell types, the authors should refrain from concluding that there is "highly similar ribosomal gene expression between HSCs and more committed progenitors". In my opinion, there could still be modest differences in ribosomal gene expression between HSCs and progenitors that cannot be revealed by the methodologies that have presently been employed.

We would like to thank the reviewers once more for their valuable comments and suggestions that have substantially improved the quality of the manuscript.

REVIEWERS' COMMENTS:

Reviewer #1 (Remarks to the Author):

The authors have further revised the manuscript related to the results on ribosomal gene expression in HSPCs and differentiated lineages. They also included additional analysis on the correlation of ribosomal gene expression between HSCs and progenitor cells based on scRNA-seq in zebrafish and human. These revisions have greatly improved the clarity of the main findings, and adequately addressed the reviewer's original comments. The rich resource of scRNA-seq data and the new insights into the regulation of hematopoiesis in zebrafish with complementary analysis from mouse and human provided by this study should stimulate interests in follow-up data analysis and functional studies.

Reviewer #2 (Remarks to the Author):

Overall this manuscript is much improved. I recommend that the manuscript be accepted, but that the authors should tone down language that risks over-interpreting the data regarding ribosomal gene expression within HSCs. It is fair to conclude that ribosomal gene expression is reduced during terminal differentiation, and that ribosomal gene expression in HSCs is more similar to progenitors than differentiated cells. In general, these observations are novel and support the conclusion that cell-type specific differences in protein synthesis are not regulated by differences in ribosomal gene expression. However, in the absence of being able to isolate and analyze purified HSCs and in the absence of spike-in controls that correct for absolute mRNA content between cell types, the authors should refrain from concluding that there is "highly similar ribosomal gene expression between HSCs and more committed progenitors". In my opinion, there could still be modest differences in ribosomal gene expression between HSCs and progenitors that cannot be revealed by the methodologies that have presently been employed.

It is true that there is a lack of cell surface markers so as to isolate pure population of zebrafish HSCs. However, using human single cells from a recent published work, we show that ribosomal gene expression is highly similar between HSCs purified using surface markers and the various progenitor populations.

Nevertheless, it is not true that we did not correct for mRNA content differences between cells; instead we used gene counts and the scran package (Lun et. al, 2016) to normalise for mRNA content differences. We did not use spike-in counts for mRNA content normalisation, since Risso et al 2014 suggested that "spike-ins are not reliable enough to be used in standard global-scaling or regression-based normalization procedures". Because of this, we did not make any changes to our previous wording.

References:

- L. Lun, A. T., Bach, K. & Marioni, J. C. Pooling across cells to normalize single-cell RNA sequencing data with many zero counts. *Genome Biol.* **17**, 75 (2016).
- Risso, D., Ngai, J., Speed, T.P. & Dudoit, S. Normalization of RNA-seq data using factor analysis of control genes or samples. *Nat. Biotechnol.* **32**, 896-902 (2014).